# Trehalose supports the growth of *Aedes aegypti* cells and modifies gene expression and dengue virus type 2 replication

Andrew D. Marten[1], Douglas P. Haslitt[1], Chad A. Martin[1], Akshitha Karthikeyan[1], Daniel H. Swanson[2], Karishma Kalera[2,3], Ulysses G. Johnson[2,3], Benjamin M. Swarts[2,3], Michael J. Conway [1*]

**1** Foundational Sciences, Central Michigan University College of Medicine, Mount Pleasant, Michigan, United States of America, **2** Department of Chemistry and Biochemistry, Central Michigan University, Mount Pleasant, Michigan, United States of America, **3** Biochemistry, Cell, and Molecular Biology Graduate Programs, Central Michigan University, Mount Pleasant, Michigan, United States of America

* michael.conway@cmich.edu

## Abstract

Trehalose is a non-reducing disaccharide that is the major sugar found in insect hemolymph fluid. Trehalose provides energy, and promotes growth, metamorphosis, stress recovery, chitin synthesis, and insect flight. Trehalase is the only enzyme responsible for the hydrolysis of trehalose, which makes it an attractive molecular target. Here we show that *Aedes aegypti* (Aag2) cells express trehalase and that they can grow on trehalose-containing cell culture media. Trehalase activity was confirmed by treating Aag2 cells with trehalase inhibitors, which inhibited conversion of trehalose to glucose and reduced cell proliferation. Cell entry of a fluorescent trehalose probe was dependent on trehalose concentration, suggesting that trehalose moves across the cell membrane via passive transport. Culturing Aag2 cells with trehalose-containing cell culture media led to significant changes in gene expression, intracellular lipids, and dengue virus replication and specific infectivity, and increased their susceptibility to trehalase inhibitors. These data describe an *in vitro* model that can be used to rapidly screen novel trehalase inhibitors and probes and underscores the importance of trehalose metabolism in *Ae. aegypti* physiology and transmission of a mosquito-borne virus.

## Author summary

Mosquito-borne viruses, such as dengue virus (DENV), pose a significant global health threat, and current control methods face challenges like pesticide resistance and limited vaccine availability. Our study explores how the sugar trehalose, a key energy source in insects, affects the physiology of *Aedes aegypti*

**Data availability statement:** FASTQ files for each sample are available in the NCBI SRA database (BioProject ID: PRJNA936836 and PRJNA882813).

**Funding:** This work was supported by start-up funds from Central Michigan University College of Medicine to MJC. BMS was supported by a grant from the National Institutes of Health (R15AI117670). The funders had no role in study design, data collection and analysis, decision to publish, or preparation of the manuscript

**Competing interests:** The authors have declared that no competing interests exist.

cells and their interaction with DENV. We found that mosquito cells grow using trehalose instead of glucose, and that trehalose uptake alters gene expression, lipid metabolism, and virus replication. Importantly, DENV-infected cells grown in trehalose-rich environments produced more infectious viral particles, suggesting that trehalose influences viral transmission. We also examined trehalase, the enzyme that breaks down trehalose, and showed that its inhibition reduces mosquito cell growth and virus replication. This highlights trehalase as a potential target for developing new insecticides or antiviral strategies. Our findings improve our understanding of mosquito metabolism and virus transmission, providing a foundation for innovative approaches to controlling *Ae. aegypti* populations and the diseases they spread.

## Introduction

*Aedes aegypti* is the primary vector of several important global human pathogens including dengue virus (DENV), Zika virus (ZIKV), and chikungunya virus (CHIKV). There are no targeted antivirals for these viruses and the currently licensed DENV vaccine is restricted to a subset of the human population due to the potential of eliciting antibody-dependent enhancement (ADE) in those who have not previously been infected [1–6]. Further, resistance to pesticides including carbamates, organochlorines, organophosphates, and pyrethroids has been detected in the Americas, Africa, and Asia. New strategies to control disease vectors are desperately needed [7–13].

Trehalose is a non-reducing disaccharide and is the most prevalent sugar found in insect hemolymph fluid. Trehalose provides energy but also promotes growth, metamorphosis, stress recovery, chitin synthesis, and insect flight [14–31]. Previous research has shown that trehalose is synthesized in the insect fat body from glucose 6-phosphate and UDP-glucose by the conserved enzymes trehalose 6-phosphate synthase (TPS) and trehalose 6-phosphate phosphatase (TPP) [19,32–34]. The hydrolysis of trehalose is under the enzymatic control of trehalase, which catalyzes the conversion of trehalose into two glucose molecules [25,34]. There are two forms of insect trehalase. Trehalase 1 (Tre-1) is soluble and has been purified from hemolymph, midgut goblet cells, and eggs. Trehalase 2 (Tre-2) is a membrane bound form and has been identified in flight muscle, follicle cells, ovary cells, spermatophore, midgut, and brain tissue [25]. Although it is known that the facilitated trehalose transporter (TRET1) transfers newly synthesized trehalose from the fat body across the cellular membrane and into the hemolymph, it is unclear if trehalose transporters exist in peripheral tissues and import this sugar [25].

Our previous research revealed that treatment of *Ae. aegypti* with the trehalase inhibitor validamycin A (ValA) promoted hypoglycemia and trehalose accumulation, and corresponded with delayed larval and pupal development, ultimately preventing flight in adult mosquitoes [26]. Unfortunately, a novel trehalose analogue called 5-ThioTre was unable to recapitulate the activity of ValA [26]. We sought to develop an *in vitro* model that would facilitate screening of novel trehalose analogues and

to determine the impact of trehalose metabolism on a mosquito-borne virus. Here, we show that the *Ae. aegypti* cell line (Aag2) expresses a secreted trehalase isoform, and that this enzyme facilitates growth when cell culture media glucose is exchanged for trehalose. Specifically, the trehalase inhibitor validamycin A (ValA) and novel trehalase inhibitors prevented conversion of cell culture media trehalose to glucose, and treatment with ValA significantly reduced cell proliferation. Cell entry of a fluorescent trehalose probe was dependent on trehalose concentration, suggesting that trehalose moves across the cell membrane via passive transport. Importantly, RNA-Seq analysis of Aag2 cells grown in trehalose-containing cell culture media revealed significant changes in gene expression in multiple cellular pathways, including those involved in integral/intrinsic membrane components. Trehalose also increased the concentration of intracellular lipids and viral RNA (vRNA), shedding of infectious virus, and the specific infectivity of DENV virions, while increasing the susceptibility of Aag2 cells and DENV to trehalase inhibitors. This manuscript describes the use of an *in vitro* model that can be used to interrogate trehalose metabolism in insect cells and reveals how trehalose modulates mosquito cell biology and virus infection.

## Methods

### Cells and reagents

The *Ae. aegypti* cell line, Aag2 (ATCC, VA), was used for *in vitro* studies. Aag2 cells were maintained at 28 °C with 5% $CO_2$ in complete cell culture media containing DMEM no glucose media supplemented with 4,500 mg/L glucose, 10% heat-inactivated fetal bovine serum (FBS) (Gemini, CA), 1% penicillin-streptomycin (Thermo Fisher Scientific, MA) and 1% tryptose phosphate broth (Sigma-Aldrich, MO). Aag2 cells were also grown using the above cell culture media with equimolar concentrations of trehalose (2,250 mg/L) (Sigma-Aldrich, MO) instead of glucose. It is important to note that the trehalose-containing cell culture media retained low concentrations of glucose from FBS and tryptose phosphate broth. The glucose concentration in our trehalose-containing cell culture media was approximately 125 mg/L but rapidly increased in glucose concentration after inoculation onto Aag2 cells. The virus strain used was dengue virus type 2 New Guinea C (DENV2 NGC). DENV2 stocks were made by infecting Aag2 cells at a MOI of 1.0 and cell-free virus stocks were harvested 8 days post-infection (dpi) and stored at −80 °C until use.

### Trehalose analogues

Validamycin A (ValA) (Cayman Chemical, MI) was the only commercially available trehalase inhibitor available at the time of this study. The synthesis of trehalose analogues was carried out using a chemoenzymatic method as previously reported for 5-ThioTre [35] and 6-TreNH$_2$ [36]. The water solubility of 5-ThioTre and 6-TreNH$_2$ is predicted to be very close to trehalose and stocks were generated in water at 100 mM.

### Synthesis of fluorescent trehalose derivative NBD-Trehalose

To a 25 mL round bottom flask flushed under argon was added anhydrous *N,N*-dimethylformamide (10 mL), 6-TreNH$_2$ (10.9 mg, 0.032 mmol), 4-chloro-7-nitrobenzo-2-oxa-1,3-diazole (7.6 mg, 0.038 mmol), and *N,N*-diisopropylethylamine (8 μL, 0.043 mmol). After stirring at room temperature under argon overnight, thin-layer chromatography indicated that the reaction was complete. The crude product was concentrated by rotary evaporation and purified by reverse phase chromatography on a Biotage Isolera One automated flash chromatography system (10 g C18 column; 0% $CH_3CN$ in $H_2O$ ◊ 80% $CH_3CN$ in $H_2O$) to give NBD-Trehalose (12.9 mg, 80%) as an orange solid. $^1$H NMR (500 MHz, $D_2O$): δ 8.38 (d, *J* = 9.0 Hz, 1 H), 6.38 (d, *J* = 9.0 Hz, 1 H), 5.18 (d, *J* = 4.0 Hz, 1 H), 5.06 (d, *J* = 3.5 Hz, 1 H), 4.11-4.07 (m, 1 H), 3.94 (bs, 1 H), 3.87-3.76 (m, 5 H), 3.71 (dd, *J* = 5.5, 12.0 Hz, 1 H), 3.67 (dd, *J* = 4.0, 10.0 Hz, 1 H), 3.47 (t, *J* = 10.0 Hz, 1 H), 3.46 (dd, *J* = 3.5, 10.0 Hz, 1 H), 3.38 (t, *J* = 10.0 Hz, 1 H). $^{13}$C NMR (125.7 MHz, $D_2O$): δ 146.46, 144.32, 143.82, 120.62, 138.70, 100.64, 93.47, 93.33, 72.58, 72.51, 71.17, 71.34, 70.93, 70.89, 70.51, 69.61, 60.49, 48.59. HRMS (ESI-TOF) m/z: [M + Na]$^+$ Calcd for $C_{18}H_{24}N_4O_{13}Na$ 527.1238; Found 527.1249. Stocks of NBD-Trehalose were generated in dimethylsulfoxide (DMSO) at 100 mM.

## Cell proliferation and glucose assays

Cell proliferation assays were performed by seeding 100 Aag2 cells into a 96 well plate in triplicate. The total number of viable cells per well were quantified using trypan blue exclusion each day starting on day 0 and ending on day 7. This assay was repeated using cells that were fed either glucose or trehalose-containing cell culture media in the presence or absence of 1 mM validamycin A (ValA). A Glucose Colorimetric Assay Kit (Cayman Chemical, MI) was first used to assay glucose concentration that developed over the course of 0–72 hours at room temperature in reactions that included 450 μL 2,250 mg/L trehalose in no glucose DMEM (Thermo Fisher Scientific, MA) and either 50 μL Aag2 cell-free supernatant from cells grown in trehalose-containing cell culture media for 3 days, 50 μL Aag2 cell-free supernatant from cells grown in trehalose-containing cell culture media for 3 days plus 1 mM ValA, or 50 μL trehalose-containing cell culture media as a negative control. Glucose Colorimetric Assay Kits were also used to quantify glucose concentration in Aag2 cell-free supernatant from cells grown in either glucose or trehalose-containing cell culture media each day starting on day 0 and ending on day 3. Glucose concentration was also assayed in Aag2 cell-free supernatant from cells grown in trehalose-containing cell culture media and in the presence of increasing concentrations of ValA, and trehalose analogues 5-ThioTre and 6-TreNH2, which are described above. Glucose concentration was also assayed in Aag2 cell-free supernatant from cells that were infected with 1 MOI DENV2 every other day starting 1-day post-infection (dpi) and ending 7 dpi.

## NBD-Tre probe assays

Aag2 cells were grown to confluence in trehalose-containing cell culture media using the Nunc Lab-Tek Chamber Slide System (Thermo Fisher Scientific, MA). Cell culture media was removed and replaced with cell culture media spiked with 0.1 mM NBD-Tre probe with and without 1 mM ValA. Negative controls included wells treated with equimolar concentrations of the DMSO vehicle control. Treated cells were incubated for 4 hours, cell culture media was then removed, and cells were stained with the nuclear stain Hoescht (Thermo Fisher Scientific, MA). Cells were imaged immediately using a Leica DM5000 B, Leica DFC550 digital camera, and Leica CTR5000 control unit. Mean fluorescence intensity (MFI) per cell was quantified using ImageJ and included 10 separate images per condition and the average MFI of 5 cells per image. This experiment was repeated by treating Aag2 cells with 0.1 mM NBD-Tre probe in the presence of increasing concentrations of glucose or trehalose and on uninfected and DENV2-infected cells.

## Nile Red assay

Aag2 cells were grown to confluence in either glucose or trehalose-containing cell culture media using the Nunc Lab-Tek Chamber Slide System. A 30 mM Nile Red (Cayman Chemical, MI) stock in DMSO was diluted 1:3000 in either glucose or trehalose-containing cell culture medium and incubated for 30 minutes at room temperature. Cells were fixed and counterstained with the nuclear stain Hoescht. Cells were imaged immediately using a Leica DM5000 B, Leica DFC550 digital camera, and Leica CTR5000 control unit. Corrected total cell fluorescence (CTCF) was quantified using integrated density, area, and mean background fluorescence data obtained from 3 separate images per condition and 20 cells per image.

## RNA-Seq

RNA Sequencing (RNASeq) was conducted using a service offered by Novogene, USA (California UC Davis) on Aag2 cells that were either maintained in glucose or trehalose-containing cell culture media in triplicate. RNA was extracted using the Qiagen RNeasy Plus RNA Extraction kit with gDNA eliminator columns. RNA integrity was assessed using the RNA Nano 6000 Assay Kit of the Bioanalyzer 2100 system (Agilent Technologies, CA, USA). Total RNA was used as input material for the RNA sample preparations. Briefly, mRNA was purified from total RNA using poly-T oligo-attached magnetic beads. Fragmentation was carried out using divalent cations under elevated temperature in First Strand Synthesis

Reaction Buffer (5X). First strand cDNA was synthesized using random hexamer primer and M-MuLV Reverse Transcriptase (RNase H-). Second strand cDNA synthesis was subsequently performed using DNA Polymerase I and RNase H. Remaining overhangs were converted into blunt ends via exonuclease/polymerase activities. After adenylation of 3' ends of DNA fragments, Adaptor with hairpin loop structure were ligated to prepare for hybridization. In order to select cDNA fragments of preferentially 370~420 bp in length, the library fragments were purified with AMPure XP system (Beckman Coulter, Beverly, USA). Then PCR was performed with Phusion High-Fidelity DNA polymerase, Universal PCR primers and Index (X) Primer. At last, PCR products were purified (AMPure XP system) and library quality was assessed on the Agilent Bioanalyzer 2100 system. The clustering of the index-coded samples was performed on a cBot Cluster Generation System using TruSeq PE Cluster Kit v3-cBot-HS (Illumia) according to the manufacturer's instructions. After cluster generation, the library preparations were sequenced on an Illumina Novaseq platform and 150 bp paired-end reads were generated. Raw data (raw reads) of fastq format were firstly processed through in-house perl scripts. In this step, clean data (clean reads) were obtained by removing reads containing adapter, reads containing ploy-N and low-quality reads from raw data. At the same time, Q20, Q30 and GC content the clean data were calculated. All the downstream analyses were based on the clean data with high quality. Reference genome and gene model annotation files were downloaded from genome website directly. Index of the reference genome was built using Hisat2 v2.0.5 and paired-end clean reads were aligned to the reference genome using Hisat2 v2.0.5. We selected Hisat2 as the mapping tool for that Hisat2 can generate a database of splice junctions based on the gene model annotation file and thus a better mapping result than other non-splice mapping tools. featureCounts v1.5.0-p3 was used to count the reads numbers mapped to each gene. FPKM of each gene was calculated based on the length of the gene and reads count mapped to this gene. FPKM, expected number of Fragments Per Kilobase of transcript sequence per Millions base pairs sequenced, considers the effect of sequencing depth and gene length for the reads count at the same time, and is currently the most commonly used method for estimating gene expression levels. Differential expression analysis of the two conditions/groups was performed using the DESeq2 R package (1.20.0). DESeq2 provide statistical routines for determining differential expression in digital gene expression data using a model based on the negative binomial distribution. The resulting P-values were adjusted using the Benjamini and Hochberg's approach for controlling the false discovery rate. Genes with an adjusted P-value <= 0.05 found by DESeq2 were assigned as differentially expressed. Functional annotations of differentially expressed genes were performed using the g:Cost Functional Profiling tool at g:Profiler. FASTQ files for each sample are available in the NCBI SRA database (BioProject ID: PRJNA936836 and PRJNA882813).

## Focus-forming unit assays

Aag2 cells were seeded into 96 well plates and grown to ~70% confluence in glucose-containing cell culture media. Wells were then inoculated with 20 focus-forming units (FFUs) of DENV2 in either glucose or trehalose-containing cell culture media for 1 hour at 27°C. Unbound virus was removed and fresh glucose-containing cell culture media was added. At 3 dpi, confluent Aag2 monolayers were fixed in 4% paraformaldehyde in phosphate buffered saline (PBS) for 10 minutes, cell membranes were permeabilized with 0.1% Triton X-100, and DENV2 antigen was stained with 1:200 anti-DENV2 envelope antibody (3H5-1, EMD Millipore) in PBS with 1% bovine serum albumin (BSA). The total number of DENV-positive foci were revealed using a 1:200 horseradish peroxidase-conjugated secondary antibody in PBS plus 1% BSA, and an AEC Peroxidase Substrate kit (Vector Laboratories). This experiment was repeated using cells that were maintained in either glucose or trehalose-containing cell culture media throughout the experiment, and digital images were collected to quantify the total focus-forming units/well and total number of DENV2-positive cells/well. FFU assays were also performed using cell free supernatants isolated from DENV2-infected Aag2 cells that were maintained in either glucose or trehalose-containing cell culture media, examining cell free supernatants at 0, 1, and 3 dpi. Cell free supernatants were stored at -80°C until use and then used to inoculate subconfluent Aag2 cell monolayers that were maintained in glucose-containing cell culture media. FFU assays were also used to quantify the impact of ValA treatment on DENV2

infection. For this experiment, Aag2 cells were inoculated with 40 FFU/well DENV2 and co-treated with 0, 0.1, 0.2, and 0.5 mg/mL ValA. Digital images were captured using the Evos XL Core Imaging System.

### RT-qPCR-based infectivity assays

Aag2 cells were seeded into 96 well plates and grown to ~70% confluence in either glucose or trehalose-containing cell culture media. Cells were then mock infected or inoculated with $1 \times 10^6$ genomic equivalents of DENV2 in either glucose or trehalose-containing cell culture media in triplicate. For the attachment assay, incubation of DENV2 with cells took place at 4°C for 1 hr. and then unbound virus was removed, cells were washed with PBS, and total RNA was extracted and quantified by RT-qPCR. DENV2 viral RNA (vRNA) was normalized by total nanograms of cellular RNA. For the cell entry and replication assay, incubation of DENV2 with cells took place at 25°C for 1 hr. and then unbound virus was removed and replaced with either glucose or trehalose-containing cell culture media. Total RNA was extracted 3 dpi, quantified by RT-qPCR, and normalized using total nanograms of cellular RNA. DENV2 stocks were also generated using Aag2 cells maintained in either glucose or trehalose-containing cell culture media, frozen at -80°C, and then RNA was extracted and quantified using RT-qPCR. FFUs were also measured in these samples, which allowed for determination of the specific infectivity of virions produced from glucose and trehalose-fed cells. Specific infectivity was represented as FFU/vRNA. Amplification of the viral target was performed using a singleplex format in 48-well plates (Illumina) with a total reaction volume of 10 µl. Reverse transcription and quantitative PCR were performed in the same closed tube with 50 ng of total RNA per reaction using the Quantitect RT-qPCR Kit (Qiagen). All primers were used at final concentrations of 4 µM. For amplification of DENV2 vRNA (nt 87–207), Forward: 5′ CAG ATC TCT GAT GAA TAA CCA ACG 3′ and Reverse: 5′ CAT TCC AAG TGA GAA TCT CTT TGT CA 3′ DENV2-specific primers were used. The DENV2-specific probe, 5′/56-FAM/CTG CCG ATC/ZEN/TTG CAC ATT ACC ACC GC/3IABkFQ/3′, was used for added specificity. All RT-qPCR reactions were performed using an Eco Illumina and each time point was measured in triplicate. Cycling conditions were 50 °C for 30 min (reverse transcription) and 95 °C for 15 min, followed by 45 cycles of 94 °C for 15 s, 55 °C for 30 s and 72 °C for 30 s. Relative quantities of target cDNA were determined using the Pfaffl method. Student's t tests were performed between groups to determine statistical significance.

## Results

### Aag2 cells express a secreted trehalase enzyme that can convert trehalose to glucose

An RNA-Seq dataset that characterized gene expression of Aag2 cells revealed high expression of trehalase (TREH; AAEL009658), but the previously predicted trehalose-6-phosphate synthase (TPS; AAEL006446) was absent in the analysis [37]. The gene that provides trehalose 6-phosphate phosphatase (TPP; AAEL010684) activity in *Ae. aegypti* was predicted by UniProtKB/TrEMBL and had high expression in Aag2 cells (Fig 1A). Based on this analysis, it was unclear if Aag2 cells could synthesize trehalose, but they could likely metabolize trehalose. WoLF PSORT analysis of the TREH amino acid sequence predicted that it is distributed across the endoplasmic reticulum, plasma, extracellular space, peroxisome, and Golgi apparatus (Fig 1B). The amino acid sequence of TREH was also analyzed using SignalP 6.0 and indicated a signal peptide cleavage site between amino acid positions 30 and 31 (probability 0.97812) (Fig 1C). Together, these data indicated that Aag2 TREH can be localized inside and outside of the cell. To confirm that a secreted form of TREH was produced by Aag2 cells, cell-free supernatant was harvested 3 days post-feeding with trehalose-containing cell culture medium and spiked into a trehalose solution for 0, 1, 12, 24, 48, and 72 hours at room temperature. Separate reactions that included 1 mM trehalase inhibitor validamycin A (ValA) in the above cell free supernatant and trehalose-containing cell culture medium that had not been in contact with Aag2s were included as negative controls. Cell free supernatant was able to increase glucose concentration nearly 4-fold, which was inhibited by ValA (Fig 1C). This supports that at least some of the TREH produced by Aag2 cells is secreted into the extracellular environment.

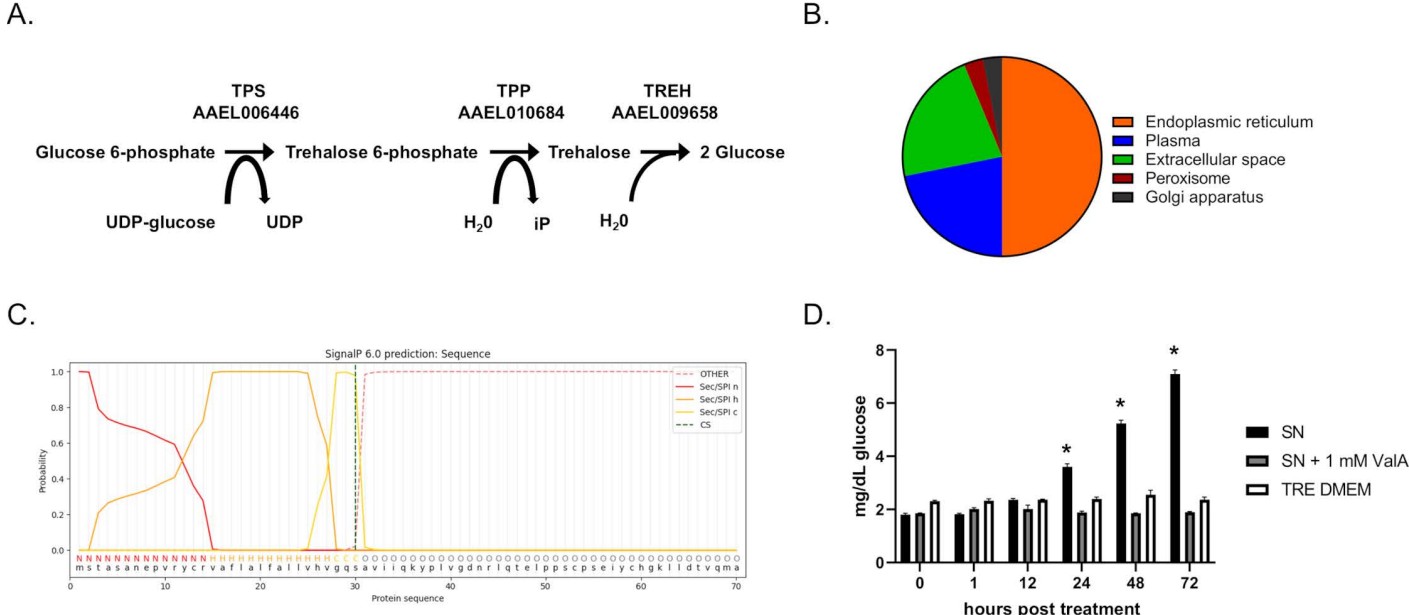

**Fig 1. Trehalose metabolism genes expressed in Aag2 cells.** (A) Trehalose pathway showing *Ae. aegypti* enzymes involved in synthesis: TPS; AAEL006446 and TPP; AAEL010684 and catabolism: TREH; AAEL009658. (B) WoLF PSORT analysis of TREH amino acid sequence predicts the cellular distribution of trehalase. (C) SignalP 6.0 analysis of TREH amino acids sequence predicts a Sec/SPI signal peptide cleavage site at S30 and associated n-terminal (n), hydrophobic (h), and c-terminal (c) regions. (D) Glucose assay of Aag2 cell free supernatant (SN), Aag2 cell free supernatant spiked with 1 mM ValA (SN + 1 mM ValA), and trehalose-containing cell culture medium (TRE DMEM) after incubation in a trehalose solution for 0, 1, 12, 24, 48, and 72 hours. Assays were performed in triplicate. Normality tests and unpaired Student's t tests were performed between groups to assess statistical significance (asterisks indicate statistical significance, * $p < 0.0001$). Standard deviations are shown.

Most cell culture media used to support insect cell growth contains glucose. We reasoned that if Aag2 expressed TREH, and if this enzyme was secreted, it would convert trehalose to glucose and similarly support cell growth. Aag2 cells were grown in both glucose and trehalose-containing cell culture media and passaged at least two times before experiments. Aag2 cells proliferated equally in glucose and trehalose-containing cell culture media (Fig 2A and 2B). Importantly, over a 3-day time course Aag2 metabolized glucose when maintained on glucose-containing cell culture media, and generated glucose when maintained on trehalose-containing cell culture media (Fig 2C and 2D).

### *In vitro* trehalase inhibitor assays show reduction in cell proliferation and reveal differences in inhibitor efficacy

Previous *in vivo* research showed that treatment with trehalase inhibitor ValA at 0.2, 0.4, and 1.0 mM led to dose-dependent changes in *Ae. aegypti* development [26]. Cell proliferation assays were performed to determine if trehalase inhibitors interfered with Aag2 cell metabolism when maintained in trehalose-containing cell culture media. There was no reduction in cell proliferation when Aag2 cells maintained in glucose-containing cell culture media were treated with 1 mM ValA (Fig 2E). However, when Aag2 cells maintained in trehalose-containing cell culture media were treated with 1 mM ValA, a significant reduction in cell proliferation was evident 3 days post-treatment, which persisted throughout the time course (Fig 2E). ValA is a competitive inhibitor of trehalase and other trehalose analogues have previously been shown to inhibit trehalase. To directly test the impact of trehalose analogues on trehalase activity, a dose-dependence assay was performed that quantified conversion of trehalose to glucose. ValA maintained ~75% activity as a trehalase inhibitor at the lowest concentration tested (i.e., 0.008 mM) (Fig 2F). Although 5-ThioTre and 6-TreNH$_2$ maintained ~75% activity at the highest concentration tested (i.e., 1.0 mM), they had no activity at the two lowest concentrations (Fig 2F).

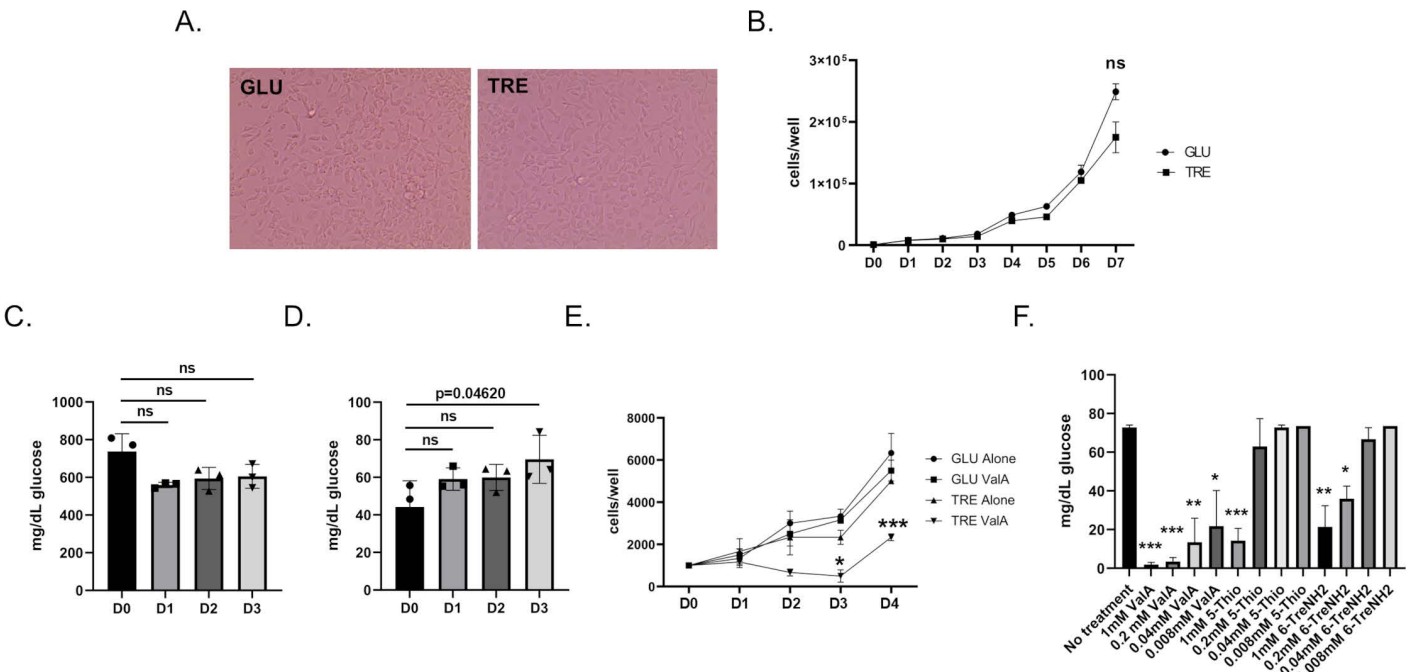

**Fig 2. Trehalose supports the growth of Aag2 cells.** (A) Representative image of Aag2 cells at 20X magnification maintained in either glucose (GLU) or trehalose (TRE)-containing cell culture media. (B) Cell proliferation assay of Aag2 cells maintained in either glucose (circles) or trehalose (squares)-containing cell culture media was performed for 7 days (D0-D7). (C) Glucose assay of Aag2 cell supernatant at days 0, 1, 2, and 3 post-feeding (D0-D3) with glucose-containing cell culture media. (D) Glucose assay of Aag2 cell supernatant at days 0, 1, 2, and 3 (D0-D3) post-feeding with trehalose-containing cell culture media. (E) Cell proliferation assay of Aag2 cells grown in either glucose-containing cell culture media with (square) or without (circle) 1.0 mM ValA or trehalose-containing cell culture media with (upside down triangle) or without (triangle) 1.0 mM ValA. (F) Glucose assay of Aag2 cell supernatants from cells maintained in trehalose-containing cell culture media and treated with 0.008-1 mM ValA, 5-ThioTre, and 6-TreNH2. Assays were performed in triplicate. Normality and unpaired Student's t tests were performed between groups to assess statistical significance (asterisks indicate statistical significance, * $p < 0.01$, ** $p < 0.001$, *** $p < 0.0001$, ns indicates non-significance). Standard deviations are shown.

## RNA-Seq analysis identifies differentially expressed genes involved in trehalose metabolism and innate immunity

Previous research confirmed that key enzymes involved in trehalose metabolism are expressed in Aag2 cells [37]. This was somewhat unexpected given that Aag2 cells do not derive from fat body tissue. Very little is known about the response of non-fat body cells and tissues to trehalose. RNA-Seq was performed to characterize the differentially expressed genes in Aag2 cells maintained in glucose and trehalose-containing cell culture media.

Significant differences in gene expression were observed. There were 157 upregulated and 118 downregulated genes in Aag2 cells that were maintained in trehalose-containing cell culture media (Fig 3A). Although TREH expression did not change in the RNA-Seq dataset (S1 Table) or in a follow-up RT-qPCR validation study (Fig 3B), facilitated trehalose transporter 1 Tret1 homologs (AAEL026825; AAEL014972) were upregulated 2.7 and 1.9-fold, respectively, in the presence of trehalose (S2 Table). Functional annotations of differentially expressed genes also showed significant changes in genes involved in cell-substrate adhesion, integral/intrinsic membrane components, metabolic pathways, drug metabolism, transmembrane receptor protein kinase activity, and enzyme-linked receptor protein signaling (Fig 3C and 3D). Significant changes in the expression of some innate immunity genes were also noted (Table 1).

## Passive uptake of a fluorescently labeled trehalose probe

Considering that Aag2 cells increase expression of Tret1 in response to trehalose, it is reasonable to expect that these cells can acquire trehalose from the extracellular environment (S1 and S2 Tables). To enable testing of whether Aag2

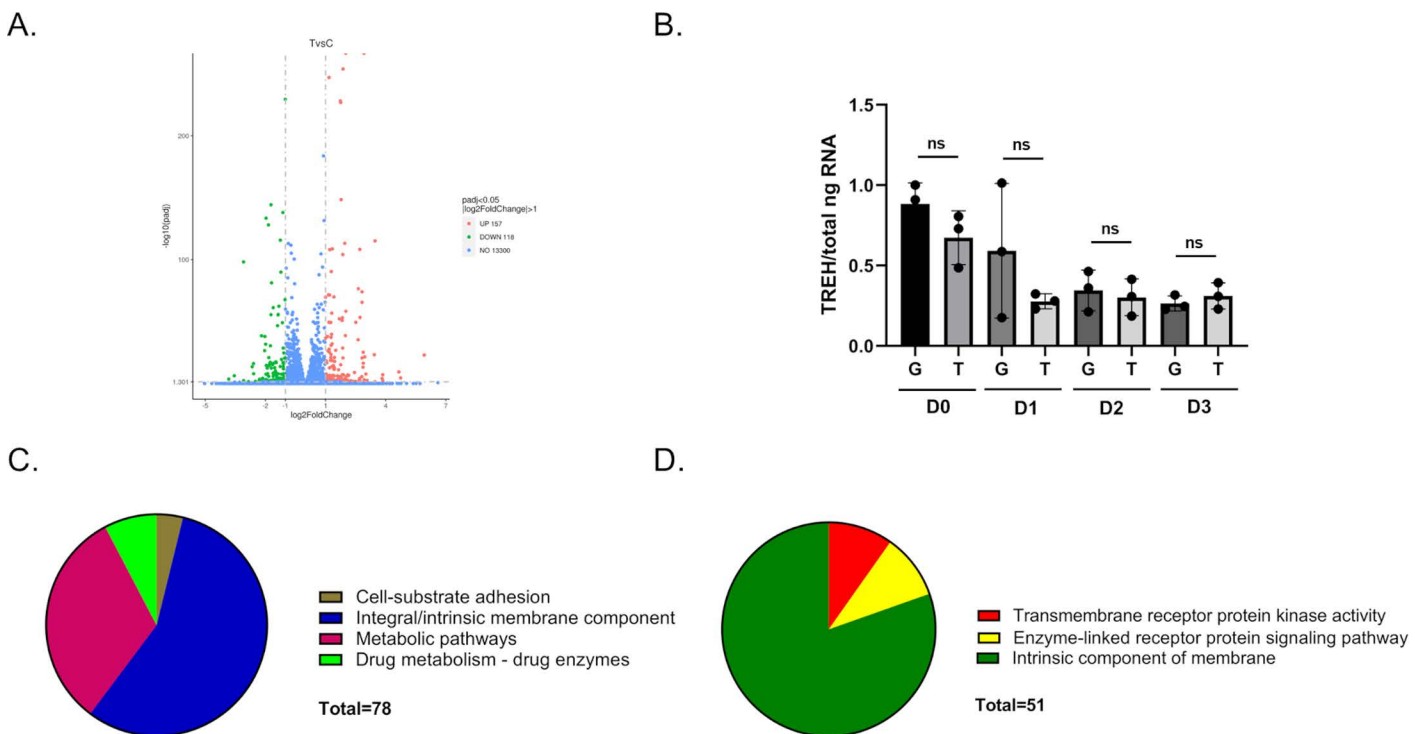

**Fig 3. RNA-Seq analysis and differentially expressed genes.** (A) Volcano plot of upregulated (157) and downregulated (118) differentially expressed genes present in Aag2 cells that were maintained in trehalose-containing cell culture media. (B) RT-qPCR analysis of TREH expression in Aag2 cells that were maintained in either glucose (G) or trehalose (T)-containing cell culture media on 0-3 dpt (D0-D3). (C) The g:Cost Functional Profiling tool at g:Profiler revealed functional annotation categories present in upregulated genes. (D) The g:Cost Functional Profiling tool at g:Profiler revealed functional annotation categories in downregulated genes. Assays were performed in triplicate. Normality and unpaired Student's t tests were performed between groups to assess statistical significance (ns indicates non-significance). Standard deviations are shown.

**Table 1. Trehalose-induced immunity genes.**

| Name | log2FoldChange | Description |
|---|---|---|
| Cat | 1.171433 | Catalase |
| AAEL009467 | 1.738973 | coactosin-like protein |
| AAEL000488 | 1.7564 | interleukin-1 receptor accessory protein-like 1 |
| N/A | 1.960177 | defensin-A-like |
| PGRPLB | 1.844781 | peptidoglycan-recognition protein |
| GRRP | 1.274502 | holotricin-3 |
| AAEL007374 | 1.327801 | protein yellow |
| AAEL009745 | 1.85381 | nitric oxide synthase |
| SRPN9 | 1.034501 | serine protease inhibitor |
| AAEL022496 | 1.013174 | peptidoglycan-recognition protein |

cells can accumulate extracellular trehalose, we synthesized a novel fluorescent trehalose derivative, NBD-Tre, which consists of trehalose conjugated to a fluorescent 2-(7-nitro-2,1,3-benzoxadiazol-4-yl) group (Fig 4A). Aag2 cells that were maintained in trehalose-containing cell culture media were treated with 1.0 mM NBD-Tre probe or a vehicle control for 4 hours, excess probe was removed, cells were washed in cell culture media, and then digital images were taken using a

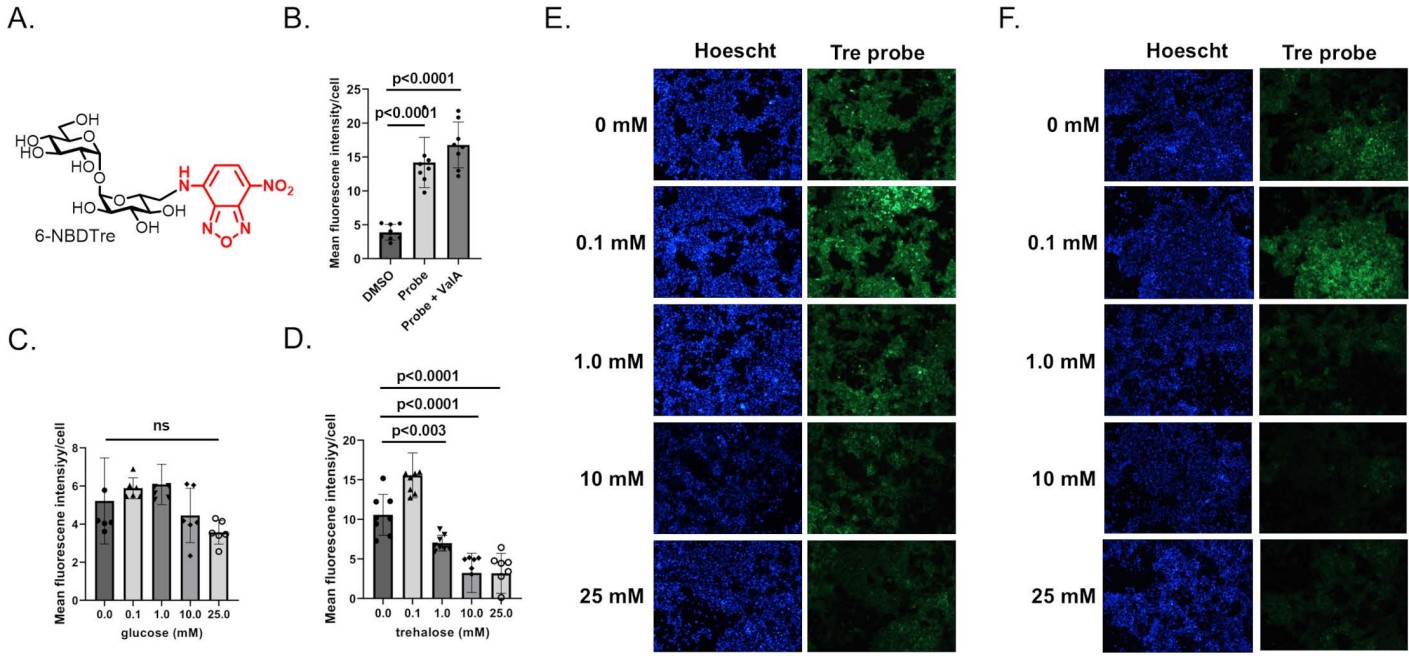

**Fig 4. NBD-Tre probe uptake in Aag2 cells.** (A) Chemical structure of fluorescently labeled trehalose probe NBD-Tre. (B) Aag2 cells were treated with either vehicle control (DMSO), 1.0 mM NBD-Tre probe, or with 1.0 mM NBD-Tre probe and 1.0 mM ValA in glucose free cell culture media for 4 hours and then mean fluorescence intensity/cell was determined. (C-D) Aag2 cells were treated with 1.0 mM NBD-Tre probe in the presence of (C) 0, 0.1, 1, 10, and 25 mM glucose and (D) 0, 0.1, 1, 10, and 25 mM trehalose for 4 hours and then mean fluorescence intensity/cell was determined. (E-F) Representative images of Aag2 cells that were treated with 1.0 mM NBD-Tre probe in the presence of (E) 0, 0.1, 1, 10, and 25 mM glucose and (F) 0, 0.1, 1, 10, and 25 mM trehalose with a Hoescht nuclear counterstain. Assays were performed in triplicate. Normality and unpaired Student's t tests were performed between groups to assess statistical significance (ns indicates non-significance). Standard deviations are shown.

fluorescent microscope. 1.0 mM ValA was used to determine if trehalase inhibition would prevent probe degradation and improve cell labeling. Quantifying the mean fluorescent intensity/cell showed a statistically significant increase in signal when cells were treated with the NBD-Tre probe compared to DMSO vehicle control. The addition of trehalase inhibitor ValA did not influence cell labeling (Fig 4B) suggesting that trehalase does not degrade the fluorescent molecule.

To test if trehalose entered into Aag2 cells using a passive/facilitated process, 0.1 mM NBD-Tre probe was suspended in cell culture media with a range of glucose and trehalose concentrations: 0.0, 0.1, 1.0, 10.0, and 25.0 mM. A lower concentration of NBD-Tre probe was used in this experiment because the labeling efficiency of 0.1 mM NBD-Tre in sugar-free cell culture media was equivalent to 1.0 mM NBD-Tre in high glucose or trehalose cell culture media. The mean fluorescent intensity per cell decreased ~10–20% at the highest two concentrations of glucose, which was statistically non-significant (Fig 4C and 4E). In contrast, the mean fluorescent intensity per cell increased at the lowest concentration of trehalose and decreased 40–75% at the three highest concentrations of trehalose (Fig 4D and 4F). These results support that the trehalose probe entered Aag2 cells through a process of facilitated diffusion, which is consistent with the function of known trehalose transporters such as Tret1 [38].

## Maintaining Aag2 cells in trehalose-containing cell culture media increases DENV2 shedding and specific infectivity

Maintaining Aag2 cells in trehalose-containing cell culture media led to significant changes in gene expression, and trehalose used a facilitated/passive process to accumulate inside of these cells. Transcriptomic changes, which include differential gene expression of key innate immunity genes (Table 1), might influence virus infection. Similarly, an influx

of trehalose into intracellular compartments may influence different aspects of the virus life cycle. Dengue virus type 2 (DENV2) was used as a model vector-borne disease to test the hypothesis that trehalose can modify virus infection in insect cells.

Aag2 cells maintained in glucose-containing cell culture media were inoculated with 20 focus-forming units (FFUs) of DENV2 and the infection was allowed to spread throughout the monolayer for 5 days. Cells were also mock-infected as a negative control. Infected and mock infected cells were then treated with either no glucose cell culture media as a vehicle control or with 1 mM NBD-Tre probe for 4 hours. Aag2 cells were then fixed and stained using an anti-DENV2 3H5.1 Mab and counterstained with Hoescht. Immunofluorescence microscopy revealed diffuse cytoplasmic localization of the NBD-Tre probe and discrete localization of DENV2 envelope (E) protein outside of the nucleus (Fig 5). There was no clear co-localization between these molecules.

Although the immunofluorescence analysis did not reveal evidence of specific co-localization between trehalose and DENV2 E, the diffuse localization of trehalose Aag2 may provide opportunities for interaction with other DENV2 proteins or to influence virus replication. We tested this hypothesis using Aag2 cells maintained in glucose-containing cell culture media, which were inoculated with 20 focus-forming units (FFUs) of DENV2 in either glucose or trehalose-containing cell culture media for 1 hour. Unbound virus and cell culture media was removed and replaced with glucose-containing cell culture media. Three days later cell monolayers were fixed and stained with anti-DENV2 MAb and the total number of FFUs per well were quantified. Pre-treatment with either glucose or trehalose-containing cell culture media had no impact

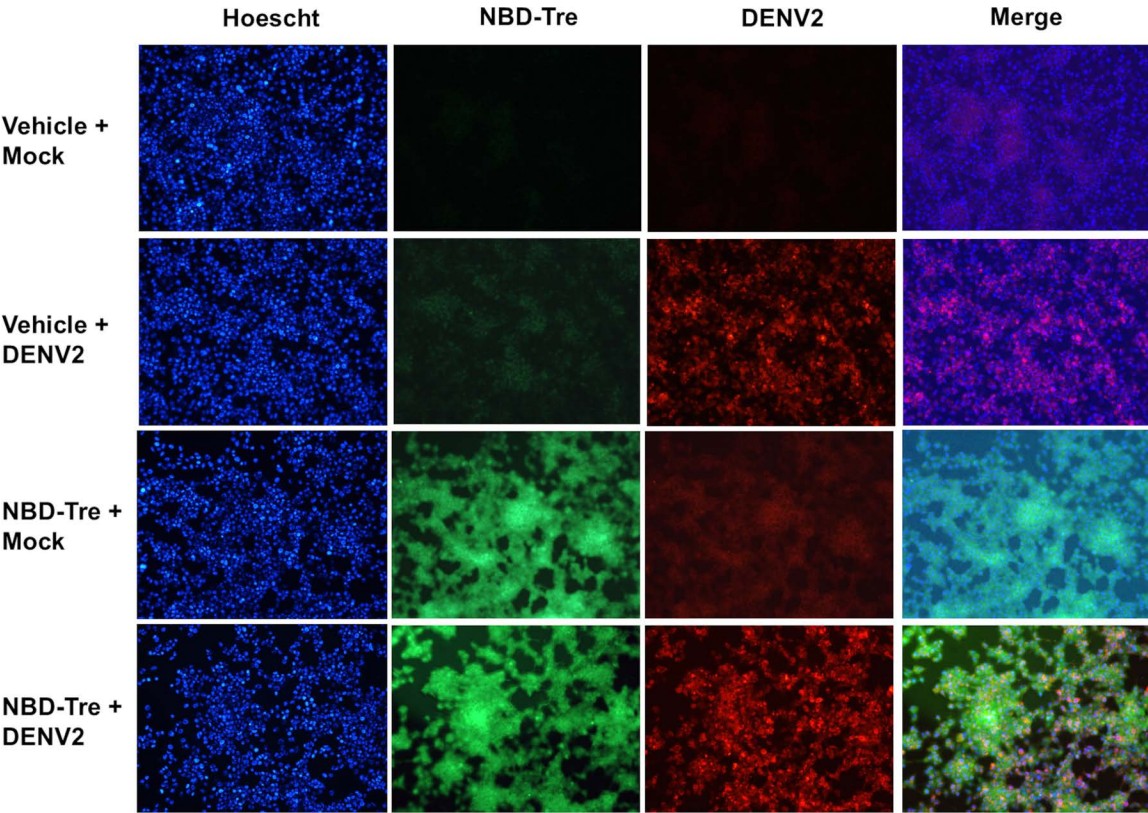

**Fig 5. Co-localization of NBD-Tre and DENV2 in Aag2 cells.** Aag2 cells were uninfected (Mock) or infected with DENV2 for 5 days and then treated with no glucose cell culture media alone (Vehicle) or 1 mM NBD-Tre probe for 4 hours. Aag2 cells were then fixed and stained with anti-DENV2 Mab and digital images were obtained at 20X magnification. Hoescht nuclear stain was added to each condition.

on DENV2 infection ([Fig 6A]). In contrast, when the same experiment was performed using Aag2 cells that were maintained in either glucose or trehalose-containing cell culture media for the duration of the focus forming unit assay, there was a 2-fold increase in FFUs per well in the trehalose-maintained cells ([Fig 6B] and [6D]). The total number of DENV2 positive cells were quantified in each well, which also revealed a 2-fold increase in the trehalose-maintained cells ([Fig 6C] and [6D]).

To determine the stage of the virus life cycle that was affected by trehalose, an attachment assay was performed using Aag2 cells that were maintained in either glucose or trehalose-containing cell culture media. Cells were inoculated with 20 FFUs DENV2 in glucose-containing cell culture media at 4 °C for 1 hour, unbound virus was removed, cells were washed with cell culture media, and total RNA was extracted from cell monolayers. RT-qPCR analysis showed no difference in the attachment of DENV2 to Aag2 cells maintained in either glucose or trehalose-containing cell culture media ([Fig 6E]). A cell entry assay was then performed, which allowed DENV2 to attach and enter Aag2 cells at 37 °C for 4 hours. RT-qPCR analysis showed a statistically significant increase in cell entry of DENV2 inside of Aag2

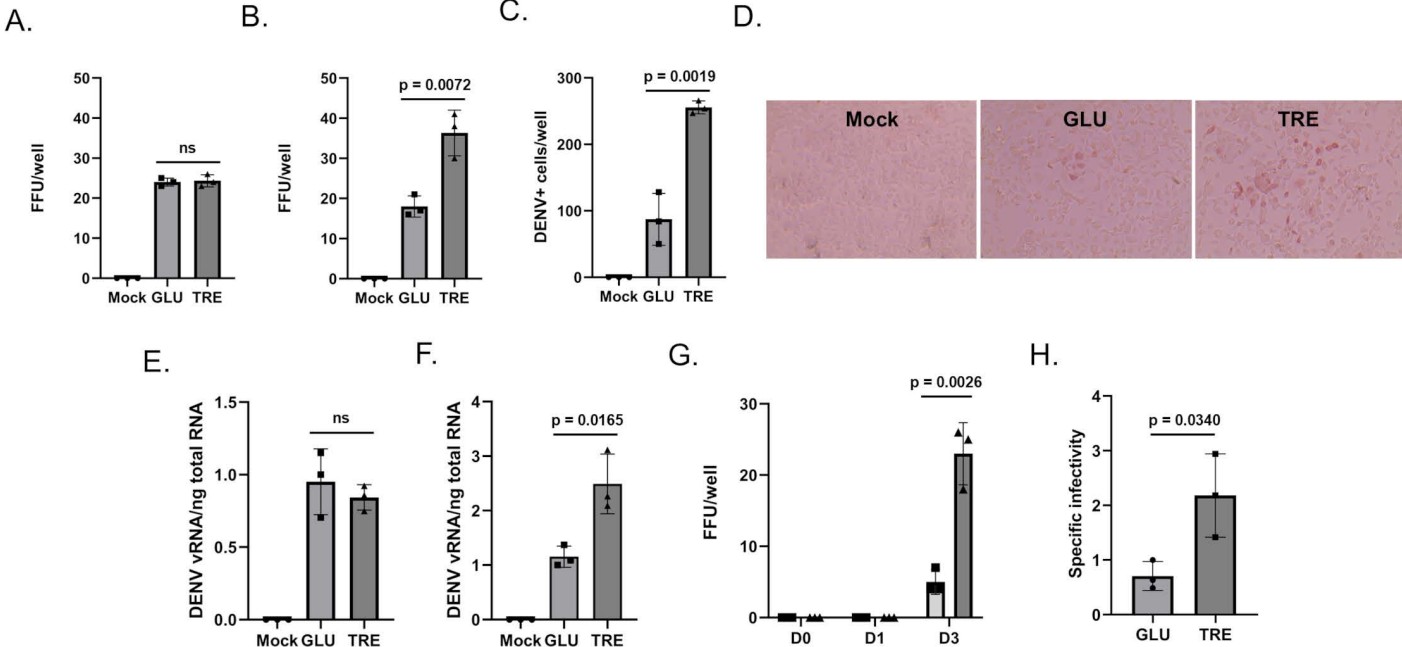

**Fig 6. Trehalose promotes DENV2 cell entry and virus shedding.** Aag2 cells were Mock infected or infected with DENV in either glucose (GLU) or trehalose (TRE)-containing cell culture media. (A) Trehalose-containing cell culture media was only present during inoculation and was then exchanged for glucose-containing cell culture media and focus-forming units were quantified 3 dpi. (B) Trehalose-containing cell culture media was present during inoculation and for the three days prior to a focus-forming unit assay. (C) The total number of DENV2 positive cells per well were quantified from (B). (D) Representative images from (B) showing distinct foci are shown. (E) An attachment assay was performed by incubating DENV2 with Aag2 cells at 4°C for 1 hour. Unbound virus was removed, cells were washed with PBS, and total RNA was extracted and quantified by RT-qPCR. DENV2 viral RNA (vRNA) was normalized by total nanograms of cellular RNA. (F) A cell entry/replication assay was performed by incubating DENV2 with Aag2 cells at 25°C for 1 hour. Unbound virus was removed and replaced with either glucose or trehalose-containing cell culture media. Total RNA was extracted 3 dpi, quantified by RT-qPCR, an DENV2 vRNA was normalized using total nanograms of cellular RNA. (G) A virus shedding assay was performed by collecting cell free supernatants at 0, 1, and 3 dpi from Aag2 cells that were inoculated with DENV2 in either glucose (squares) or trehalose-containing (triangles) cell culture media. Cell free supernatants were used to inoculate monolayers of Aag2 cells grown in glucose-containing cell culture media and focus forming unit assays were used to quantify the infectious units present in each sample. (H) DENV2 stocks were generated from Aag2 cells that were grown in either glucose or trehalose-containing cell culture media. Focus forming unit assays were performed to quantify the infectivity of each stock. Total RNA was also extracted from virus stocks and DENV2 vRNA was quantified using RT-qPCR. Specific infectivity was derived by normalizing the total number of focus forming units in a sample by its vRNA content (i.e., FFU/vRNA). Assays were performed in triplicate. Normality and unpaired Student's t tests were performed between groups to assess statistical significance (ns indicates non-significance). Standard deviations are shown.

cells that were maintained in trehalose-containing cell culture media (Fig 6F). Supernatants were then collected from DENV2-infected Aag2 cells that were maintained in either glucose or trehalose-containing cell culture media and stored as viral stocks. Viral stocks were tested to determine the total FFUs/well in a 5 µL inoculum onto Aag2 cells that were maintained in glucose-containing cell culture media. Consistent with previous experiments, significantly more DENV2 was shed from Aag2 cells that were maintained in trehalose-containing cell culture media by day 3 post-infection (dpi) (Fig 6G). To test the quality of virions produced by Aag2 cells maintained in either glucose or trehalose-containing cell culture media, virus stocks were prepared from each condition at 5 dpi, and then FFU and RT-qPCR assays were performed. The total FFUs per mL were normalized by vRNA (FFU/vRNA) to determine specific infectivity of virus stocks. The specific infectivity of DENV2 produced from Aag2 cells maintained in trehalose-containing cell culture media was 2-fold greater (Fig 6H).

### Trehalose promotes accumulation of intracellular lipids

Previous research suggests that trehalose induces autophagy in mammalian cells, although autophagy's role in virus infection is varied [39–45]. The consensus is that DENV benefits from autophagy in mammalian cells by enhancing cell survival and promoting virus maturation [46–50]. The role of autophagy in arbovirus infection in insect cells is less clear, and conventional autophagy inducers and inhibitors may respond differently in insect cells [45,51]. Importantly, there were no autophagy-specific genes or pathways that were significantly upregulated in the RNA-Seq dataset. Trehalose can also be converted to glucose and used as a substrate for lipogenesis [37].

RNA-Seq analysis revealed significant changes in gene expression related to integral/intrinsic membrane components, and the main function of autophagy is to accumulate and utilize lipids to enhance cell survival. We used Nile Red staining to determine if trehalose modified the lipid environment of Aag2 cells. We identified a slight, yet significant increase in intracellular lipid accumulation in Aag2 cells that were maintained in trehalose-containing cell culture media (Fig 7A and 7B).

### DENV2 infection does not impact trehalose metabolism

Previous research has shown that DENV manipulates the metabolism of its host cell to promote replication. Different parameters of trehalose metabolism were tested during DENV2 infection to determine if this pathway is manipulated in Aag2 cells. Aag2 cells maintained in trehalose-containing cell culture media were either mock infected or infected with DENV2 and conversion of trehalose to glucose was assessed on 1, 3, 5, and 7 dpi. DENV2 infection did not influence trehalase activity (Fig 8A). The impact of DENV2 infection on trehalose uptake was also tested by infecting Aag2 cells that were maintained in glucose-containing cell culture media with 0, 1, 10, 100, and 500 FFUs and then labeling cells with 0.1 mM NBD-Tre 3 dpi. There was no difference in mean fluorescence intensity per cell, therefore DENV2 infection did not influence trehalose uptake (Fig 8B).

### Trehalase inhibition reduces cell viability and DENV infection in trehalose-dependent Aag2 cells

We previously showed that trehalase inhibition reduced cell proliferation when cells were maintained in trehalose-containing cell culture media (Fig 2E). We hypothesized that this activity would reduce DENV productivity. As expected, ValA treatment did not reduce cell viability or DENV productivity in Aag2 cells that were maintained in glucose-containing cell culture media, and surprisingly increased DENV productivity by 50% (Fig 9A-9C). In contrast, ValA treatment significantly reduced cell viability and DENV productivity in Aag2 cells that were maintained in trehalose-containing cell culture media (Fig 9D-9F). It is important to note that a FFU assay was not performed directly on Aag2 cells that were maintained on trehalose-containing cell culture media because the addition of ValA prevents establishment of a monolayer, which is required for the assay. Instead, cell-free supernatants were harvested from ValA-treated cells and used for downstream FFU assays performed on Aag2 cells that were maintained on glucose-containing cell culture media.

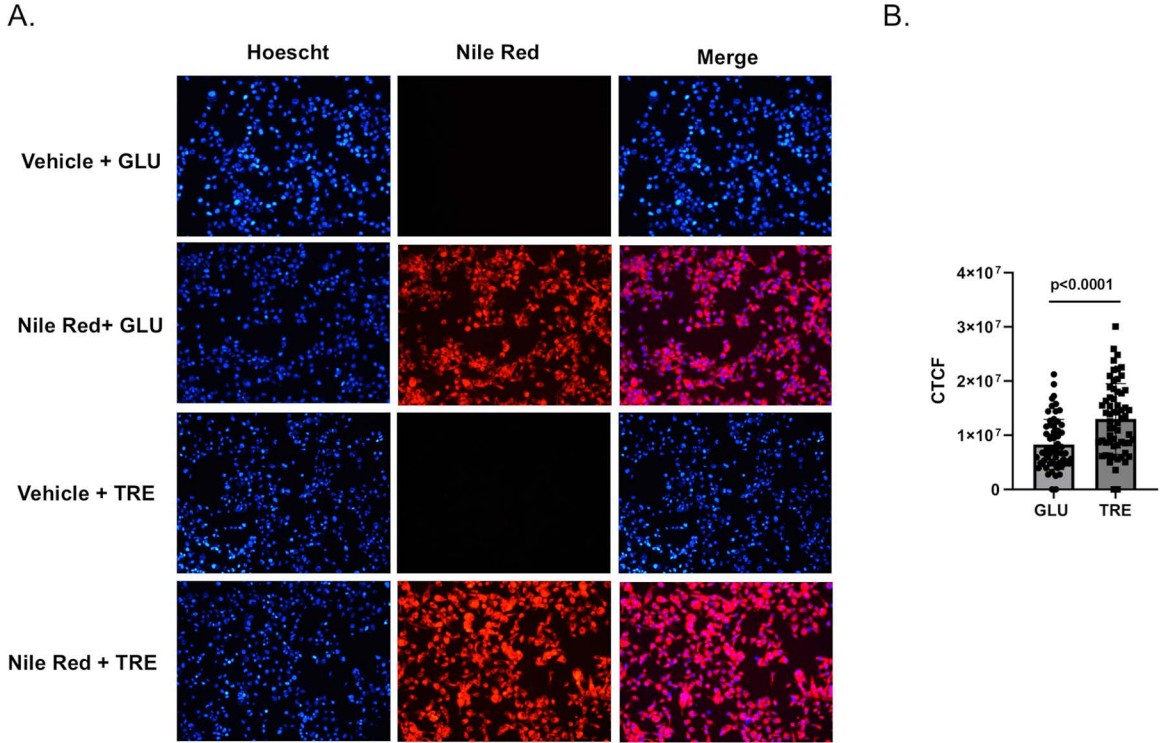

**Fig 7. Trehalose promotes intracellular lipid accumulation in Aag2 cells.** (A) Intracellular lipids were labeled in Aag2 cells maintained in either glucose (GLU) or trehalose (TRE) cell culture media using either vehicle control (DMSO) or Nile Red. Cells were fixed and digital images were obtained at 20X magnification. Hoescht nuclear stain was added to each condition. (B) The corrected total cell fluorescence (CTCF) was quantified using the integrated density of 60 individual cells per condition. Assays were performed in triplicate. Normality and unpaired Student's t tests were performed between groups to assess statistical significance. Standard deviations are shown.

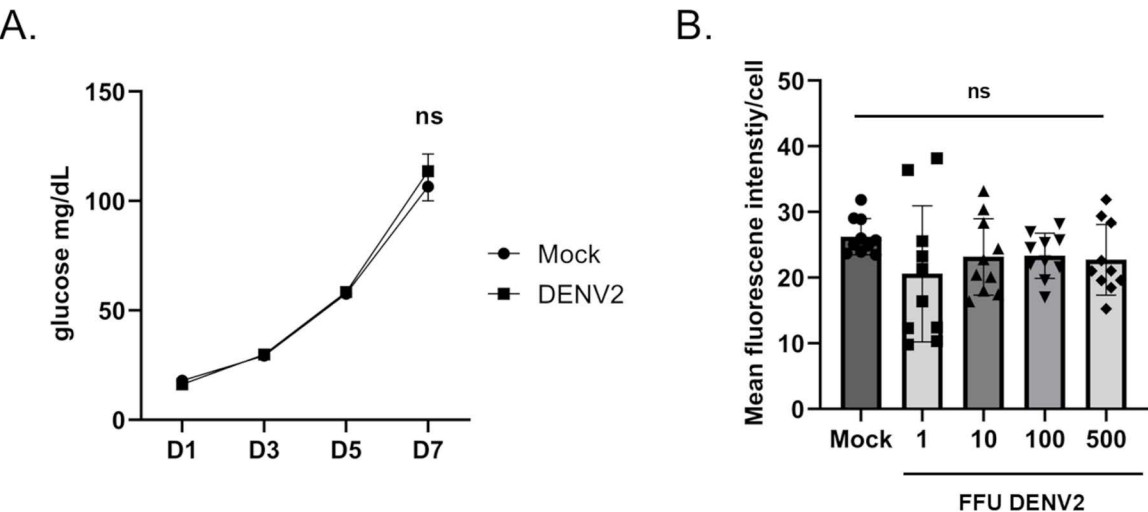

**Fig 8. DENV2 infection does not influence trehalase activity or trehalose uptake.** (A) A glucose assay was performed on cell-free supernatants taken from Aag2 cells maintained in trehalose-containing cell culture media that were either uninfected (Mock) or infected with DENV2 for 1, 3, 5, and 7 dpi. (B) A trehalose probe uptake assay was performed on Aag2 cells that were either uninfected (Mock) or infected with 1, 10, 100, and 500 FFUs of DENV2 5 dpi. 1.0 mM NBD-Tre probe was added to cells in glucose free cell culture media for 4 hours. Unbound probe was removed, and then digital images were obtained at 20X magnification. Mean fluorescence intensity was determined for 50 individual cells per condition. Assays were performed in triplicate. Normality and unpaired Student's t tests were performed between groups to assess statistical significance (ns indicates non-significance). Standard deviations are shown.

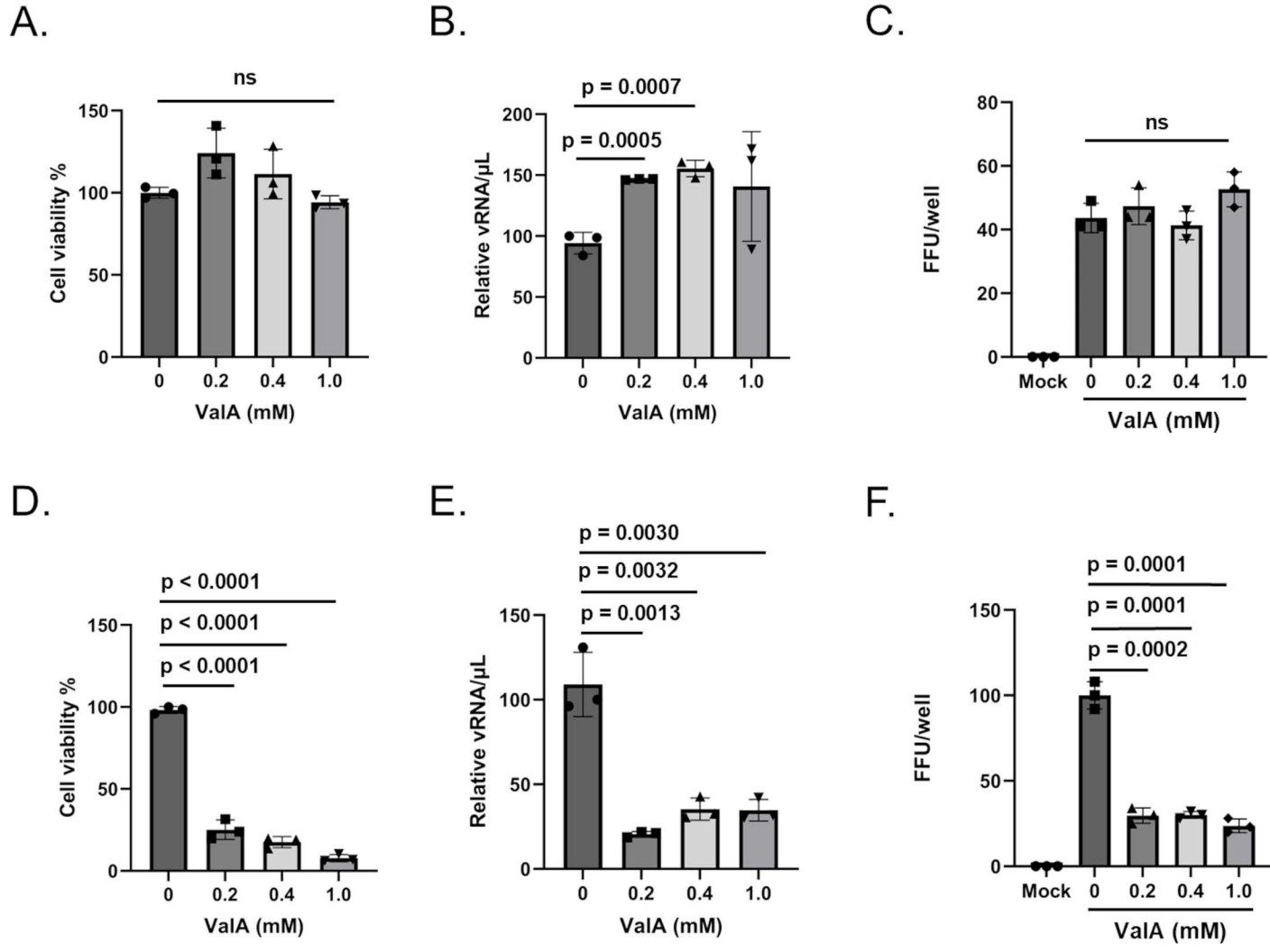

**Fig 9. Validamycin A reduces cell viability and DENV2 productivity when Aag2 cells are maintained in trehalose-containing cell culture media.**
(A) Subconfluent Aag2 cells maintained in glucose-containing cell culture media were treated with 0, 0.2, 0.4, and 1.0 mM ValA for 5 days and then trypan blue exclusion assays were performed to quantify percent cell viability. (B) Subconfluent Aag2 cells maintained in glucose-containing cell culture media were inoculated with 20 FFUs of DENV for 1 hour, followed by treatment with 0, 0.2, 0.4, and 1.0 mM ValA for 5 days. vRNA was extracted from equivalent volumes of cell free supernatants and quantified by RT-qPCR. (C) Subconfluent Aag2 cells maintained in glucose-containing cell culture media were inoculated with 20 FFUs of DENV for 1 hour, followed by treatment with 0, 0.2, 0.4, and 1.0 mM ValA for 3 days. FFU assays were performed to quantify the total FFUs present in each well. (D) Subconfluent Aag2 cells maintained in trehalose-containing cell culture media were treated with 0, 0.2, 0.4, and 1.0 mM ValA for 5 days and then trypan blue exclusion assays were performed to quantify percent cell viability. (E) Subconfluent Aag2 cells maintained in trehalose-containing cell culture media were inoculated with 20 FFUs of DENV for 1 hour, followed by treatment with 0, 0.2, 0.4, and 1.0 mM ValA for 5 days. vRNA was extracted from equivalent volumes of cell free supernatants and quantified by RT-qPCR. (F) Subconfluent Aag2 cells maintained in trehalose-containing cell culture media were inoculated with 20 FFUs of DENV for 1 hour, followed by treatment with 0, 0.2, 0.4, and 1.0 mM ValA for 3 days. Cell free supernatants were harvested and used for a downstream FFU assay performed on Aag2 cells maintained on glucose-containing cell culture media, which quantified the total FFUs present in each well. Assays were performed in triplicate. Normality and unpaired Student's t tests were performed between groups to assess statistical significance (ns indicates non-significance). Standard deviations are shown.

Since glucose and trehalose-containing cell culture media contain 10% fetal bovine serum (FBS) and 1% tryptose phosphate solution, glucose and free fatty acids (FFAs) are present in the base media. This means that ValA treatment can only prevent utilization of glucose derived from supplemental trehalose. It was possible that ValA-mediated toxicity

was not due to hypoglycemia but an unknown trehalose-dependent mechanism of action. To test this possibility, cell proliferation assays were performed using Aag2 cells maintained in either glucose-containing cell culture media or no glucose-containing cell culture media. The no glucose-containing cell culture media was made using no glucose DMEM that was supplemented with 10% FBS, 1% tryptose phosphate solution, and 1% penicillin-streptomycin solution. There was no difference in cell proliferation during a 7-day time course (Fig 10A). Similarly, a cell viability assay using 100, 50, 10, and 0 percent glucose-containing cell culture media showed no difference between groups (Fig 10B). DENV shedding and infectivity were also not impacted by the different types of cell culture media (Fig 10C and 10D). These data reveal that Aag2 cells don't require supplemental glucose present in high glucose DMEM and can grow using the glucose and FFAs present in FBS and tryptose phosphate broth. It also suggests that, while ValA-mediated toxicity is trehalose-dependent, it is not due to hypoglycemia.

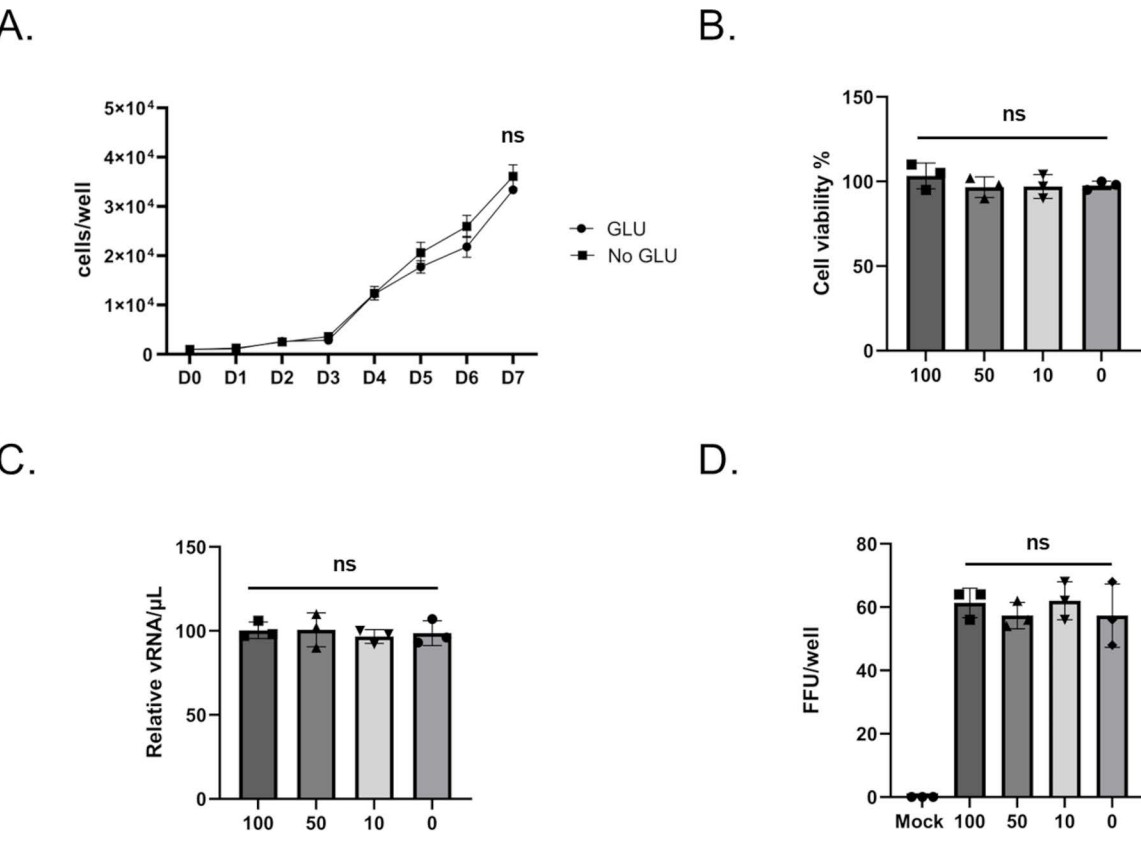

**Fig 10. Supplemental glucose in high glucose DMEM is not required for Aag2 cell proliferation, viability, or DENV shedding or infectivity.** (A) Cell proliferation assay of Aag2 cells maintained in either glucose (circles) or no glucose (squares)-containing cell culture media was performed for 7 days (D0-D7). (B) Subconfluent Aag2 cells were maintained in decreasing concentrations of supplemental glucose-containing cell culture media (100, 50, 10, and 0 percent supplemental glucose) for 5 days and then trypan blue exclusion assays were performed to quantify percent cell viability. (C) Subconfluent Aag2 cells maintained in glucose-containing cell culture media were inoculated with 20 FFUs of DENV for 1 hour, followed by treatment with decreasing concentrations of supplemental glucose-containing cell culture media (100, 50, 10, and 0 percent supplemental glucose) for 5 days. vRNA was extracted from equivalent volumes of cell free supernatants and quantified by RT-qPCR. (D) Subconfluent Aag2 cells maintained in glucose-containing cell culture media were inoculated with 20 FFUs of DENV for 1 hour, followed by treatment with decreasing concentrations of supplemental glucose-containing cell culture media (100, 50, 10, and 0 percent supplemental glucose) for 5 days. Cell free supernatants were harvested and used for a downstream FFU assay performed on Aag2 cells maintained on glucose-containing cell culture media, which quantified the total FFUs present in each well. Assays were performed in triplicate. Normality and unpaired Student's t tests were performed between groups to assess statistical significance (ns indicates non-significance). Standard deviations are shown.

## Discussion

Control of *Aedes* spp. disease vectors is an important area of ongoing research due to concerns regarding vaccine limitations, pesticide resistance, and increased geographic spread [6]. Trehalase inhibitors have been previously characterized as a novel class of putative insecticides, potentially effective for controlling *Aedes* spp. disease vectors due to the vital importance of trehalose as a blood sugar during mosquito development [6,16–18,20–28,31,34,35,52]. Our previous research has revealed that the commercially available trehalase inhibitor ValA negatively impacts mosquito development *in vivo* when applied directly to the larval environment by showing dose-dependent activity delaying egg hatch, pupation, eclosure, and flight of adult *Ae. Aegypti* mosquitoes [26].

Here, we describe the use of an *in vitro* model to screen for trehalase inhibitors in *Ae. aegypti* and determine the impact of trehalose metabolism on a mosquito-borne virus. ValA and novel trehalase inhibitors (5-ThioTre, 6-TreNH$_2$) prevented conversion of cell culture media trehalose to glucose. ValA had the strongest activity, which significantly reduced cell proliferation. Differences in the activity of trehalose analogues seen *in vitro* and *in vivo* suggest that there are opportunities to optimize trehalase inhibitors for target species [6]. Our *in vitro* assay is a platform that could be used for high throughput studies to develop targeted trehalose analogues.

RNA-Seq analysis showed that Tret1 expression was induced in cells that were maintained in trehalose-containing cell culture media. The Aag2 cell line was originally generated from whole homogenized embryos. Cells within the culture exhibit differing morphologies, and it has been suggested that the varying morphologies of mosquito cells in culture may be indicative of the presence of a diversity of embryonic and differentiated cell types [53]. Aag2 cells were not derived from fat body cells or tissue, which is where Tret1 is concentrated *in vivo* [38]. The role of Tret1 in peripheral cells and tissues is unknown. Considering that the machinery for trehalose synthesis was lacking in Aag2 cells, we hypothesized that these cells may use Tret1 for trehalose uptake rather than excretion. Previous research has shown that Tret1 is a facilitated/passive transporter [38]. We confirmed that Aag2 cells could accumulate a novel fluorescent trehalose probe, NBD-Tre, and that uptake was prevented if unlabeled trehalose was present. Increasing the concentration of glucose was less able to prevent uptake of NBD-Tre suggesting that there may have been competition at the level of sugar-specific transporters. This supports facilitated/passive uptake of trehalose in Aag2 cells.

Entry of trehalose into Aag2 cells may have led to differential gene expression as noted in the RNA-Seq dataset, although the presence of trehalose also induced expression of integrin proteins, which mediate signaling from the cell surface. Significant differences in gene expression were present in genes involved in composition of the cell membrane, lipid biosynthesis, sugar conjugation, extracellular matrix, calcium signaling, signal transduction, and mitochondrial function. These global changes in gene expression may explain why the physiological role of trehalose extends beyond a simple source of fuel. Trehalase inhibition can impact a wide array of physiological functions in insects including chitin synthesis, stress protection, larval and pupal development [14,15,17,18,20–23,25–29,31,52,54–57].

Considering that trehalose enters Aag2 cells and leads to significant changes in gene expression, we hypothesized that the environment of a trehalose-fed insect cell was different enough to influence virus infection. This was supported by previous data suggesting that trehalose can induce autophagy and that DENV2 benefits from autophagy by prolonging cell survival and enhancing virus maturation [46–50]. We found that Aag2 cells that were maintained in trehalose-containing cell culture media promoted DENV2 cell entry, which led to a higher number of infected cells, and increased virus shedding. In mammalian cells, trehalose is an inducer of autophagy and can block autophagic flux from autophagosomes to autolysosomes [44]. DENV2 also induces the autophagy pathway, and infection is reduced when autophagy is inhibited [46–50]. It is possible that trehalose increases DENV2 infection in Aag2 cells by promoting autophagy and enhancing virus cell entry through modification of cell membranes. Previous research suggests that the relationship between autophagy and DENV infection in Aag2 cells is complicated and that conventional autophagy inhibitors may not work as expected in these cells [45]. Instead, we used Nile Red staining

PLOS Pathogens

as a proxy to quantify changes to the intracellular lipid environment. Nile Red staining revealed an increase in intracellular lipids in Aag2 cells maintained on trehalose-containing cell culture media. It is important to note that trehalose is the main blood sugar in insects, and it may not have the same influence on autophagy as seen in mammalian cells.

Trehalose can also be converted to glucose, which is a substrate for lipogenesis. It is possible that trehalose stimulates lipogenesis and the production of triglycerides and associated fatty acids that promote virus replication and synthesis of more infectious viral particles. RNA-Seq analysis showed that trehalose-fed cells had a 55% downregulation of phosphoenolpyruvate carboxykinase (i.e., AAEL000006) and a 20% upregulation of fatty acid synthase (i.e., AAEL001194) (S1 Table). Further, one of the most upregulated genes (714%) was elongation of very long chain fatty acids protein 4 (AAEL004947). Together, these changes in gene expression may have been responsible for the increase in intracellular lipids and perhaps the increase in virus replication and shedding.

The specific infectivity of DENV2 also increased when virus stocks were produced from Aag2 cells that were maintained in trehalose-containing cell culture media. The mechanism of action that led to increased specific infectivity is unknown, although trehalose is widely known for its ability to prevent heat and cold shock and it is used as a cryoprotectant [55,56,58–62]. Trehalose may provide some membrane or protein stabilizing activity that either allows virus production and shedding to occur more efficiently or it simply allows more virus to survive the freeze-thaw event that occurs after removing a virus stock from the -80°C freezer prior to experimentation. Autophagy has also been linked to DENV maturation, which suggests that trehalose may facilitate production of more mature virions [50]. It is important to note that the trehalose concentration used in our cell culture media is 20–50-fold less than what is typically used for cryoprotection [58,60,63–65].

Trehalase inhibition interferes with multiple physiological pathways in insects, and we found that ValA reduced cell proliferation in Aag2 cells that were maintained in trehalose-containing cell culture media. It was reasonable to hypothesize that trehalase inhibition would also reduce DENV productivity. We showed that 1.0 mM ValA causes a 90% reduction in cell viability 5 days post-treatment in Aag2 cells that were maintained in trehalose-containing cell culture media. The reduction in cell viability correlated with an approximately 75% decrease in DENV vRNA shedding. *In vivo* studies show deficits in *Ae. aegypti* egg hatch rate, larval development, pupation, and eclosure [26]. Considering that vertical transmission is less important for the maintenance and transmission of DENV in a natural environment, trehalase inhibition in larval habitats would likely not impact DENV in a meaningful way. It is unclear how trehalose inhibition impacts adult *Ae. aegypti* at this time, although these data suggest that targeting adults could simultaneously reduce mosquito and virus fitness. Interestingly, Aag2 cells were able to grow in the absence of supplemental glucose, which suggested that ValA-mediated toxicity was not due to restriction of glucose from these cells. Both glucose and trehalose-containing cell culture media had the same concentration of glucose and free fatty acids (FFAs) in their base media due to supplementation with 10% FBS and 1% tryptose phosphate broth. That said, ValA-mediated toxicity was trehalose-dependent. It is possible that in the absence of a functional trehalase enzyme and in the presence of excess trehalose that trehalose can be toxic. This could partially be due to the ability of trehalose to enter Aag2 cells, which may lead to osmotic disturbances. It's also possible that upregulation of facilitated trehalose transporters during maintenance of cells in trehalose-containing cell culture media leads to uptake of ValA, which may have off-target effects inside of the cell. This observation further supports that trehalose acts as a functional molecule since Aag2 cells didn't require the supplemental glucose or trehalose for growth, yet addition of trehalose led to significant transcriptomic changes and infection outcomes.

This manuscript describes an *in vitro* model that can be used to rapidly screen novel trehalase inhibitors and probes. It underscores the importance of trehalose metabolism in *Ae. aegypti* physiology and the transmission of a mosquito-borne virus. These data facilitate the future development of targeted trehalose analogues that will reduce populations of disease vectors and related viruses with minimal off-target effects.

## Supporting information

**S1 Table. Complete readcount and FPKM list of all genes in GLU vs TRE Aag2 cells.**
(CSV)

**S2 Table. Complete readcount and FPKM list of upregulated genes in GLU vs TRE Aag2 cells.**
(CSV)

**S3 Table. Complete readcount and FPKM list of all downregulated in GLU vs TRE Aag2 cells.**
(CSV)

## Author contributions

**Conceptualization:** Andrew D. Marten, Benjamin M. Swarts, Michael J. Conway.

**Data curation:** Andrew D. Marten, Douglas P. Haslitt, Akshitha Karthikeyan, Michael J. Conway.

**Formal analysis:** Andrew D. Marten, Benjamin M. Swarts, Michael J. Conway.

**Funding acquisition:** Benjamin M. Swarts, Michael J. Conway.

**Investigation:** Andrew D. Marten, Douglas P. Haslitt, Chad A. Martin, Akshitha Karthikeyan, Michael J. Conway.

**Methodology:** Andrew D. Marten, Daniel H. Swanson, Karishma Kalera, Ulysses G. Johnson, Benjamin M. Swarts, Michael J. Conway.

**Project administration:** Michael J. Conway.

**Resources:** Daniel H. Swanson, Karishma Kalera, Ulysses G. Johnson, Benjamin M. Swarts, Michael J. Conway.

**Supervision:** Benjamin M. Swarts, Michael J. Conway.

**Validation:** Michael J. Conway.

**Visualization:** Michael J. Conway.

**Writing – original draft:** Michael J. Conway.

**Writing – review & editing:** Andrew D. Marten, Benjamin M. Swarts, Michael J. Conway.

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
