## [Decision Letter · Decision Letter 0]

20 Feb 2025

PPATHOGENS-D-24-02636

Trehalose supports the growth of Aedes aegypti cells and modifies gene expression and dengue virus replication

PLOS Pathogens

Dear Dr. Conway,

Thank you for submitting your manuscript to PLOS Pathogens. After careful consideration, we feel that it has merit but does not fully meet PLOS Pathogens's publication criteria as it currently stands. Therefore, we invite you to submit a revised version of the manuscript that addresses the points raised during the review process.

Please submit your revised manuscript within 60 days Apr 21 2025 11:59PM. If you will need more time than this to complete your revisions, please reply to this message or contact the journal office at plospathogens@plos.org. Please include the following items when submitting your revised manuscript:

We look forward to receiving your revised manuscript.

Kind regards,

Alan G. Goodman

Academic Editor

PLOS Pathogens

Alexander Gorbalenya

Section Editor

PLOS Pathogens

 Sumita Bhaduri-McIntosh

Editor-in-Chief

PLOS Pathogens

orcid.org/0000-0003-2946-9497

Michael Malim

Editor-in-Chief

PLOS Pathogens

orcid.org/0000-0002-7699-2064

**Additional Editor Comments:**

Thank you for your patience as it took longer than expected to gather the reviews. The reviewers understand the importance of this study, but require additional revisions before the manuscript can be considered further. Notably, they ask for evidence that trehalose is secreted and an experiment measuring viral replication in the presence of glucose.

For sake of accuracy, please consider specifying type 2 of the virus in the title and abstract.  

**Journal Requirements:**

At this stage, the following Authors/Authors require contributions: Michael J. Conway, Andrew D. Marten, Douglas P. Haslitt, Chad A. Martin, Daniel H. Swanson, Karishma Kalera, Ulysses G. Johnson, and Benjamin M. Swarts. Please ensure that the full contributions of each author are acknowledged in the "Add/Edit/Remove Authors" section of our submission form.

https://journals.plos.org/plospathogens/s/submission-guidelines#loc-parts-of-a-submission

4) Tables should not be uploaded as individual files. Please remove these files and include the Tables in your manuscript file as editable, cell-based objects. For more information about how to format tables, see our guidelines:

https://journals.plos.org/plospathogens/s/tables

5) Please amend your detailed Financial Disclosure statement. This is published with the article. It must therefore be completed in full sentences and contain the exact wording you wish to be published. State what role the funders took in the study. If the funders had no role in your study, please state: "The funders had no role in study design, data collection and analysis, decision to publish, or preparation of the manuscript.".

**Reviewers' Comments:**

Reviewer's Responses to Questions

**Part I - Summary**

Reviewer #1: In this manuscript, the authors present a novel method utilizing Aedes aegypti (Aag2) cells to evaluate trehalase inhibitors, offering a valuable platform for screening new compounds targeting trehalase. This is significant given the lack of antiviral treatments for arboviruses and the limited availability of vaccines due to the risk of antibody-dependent enhancement (ADE). Previously, the authors demonstrated that trehalase inhibitors induce hypoglycemia in vivo, delay mosquito development, and impair adult flight capacity, highlighting their potential as a new class of insecticides. Trehalase, the sole enzyme hydrolyzing trehalose into glucose, plays a central role in insect hemolymph metabolism, making reliable tools to assess trehalase inhibitors in Aedes aegypti cells highly valuable. Additionally, this study examines whether trehalose influences arbovirus infection in Aedes aegypti cells, providing both methodological advancements and insights into mosquito metabolism with implications for arbovirus control.

The authors reveal that Aag2 cells actively secrete functional trehalase, supported by bioinformatics and the conversion of trehalose into glucose. Introducing ValA, a trehalase inhibitor, significantly reduced Aag2 cell proliferation, mirroring previous in vivo observations. They also established an in vitro system to test additional inhibitors, confirming its reliability as other compounds similarly suppressed glucose production at high concentrations. Addressing the unexplored relationship between trehalose metabolism and arbovirus infection, RNA-seq analysis showed TRET1, a putative trehalose transporter, upregulated in response to trehalose, while TREH and TPP remained unchanged. This suggested TRET1's role in trehalose uptake, further supported by gradient-dependent probe experiments. Trehalose was shown to passively enter cells, implicating specific transport mechanisms, though TRET1's exact role remains unclear. In exploring trehalose’s impact on dengue virus (DENV) infection, trehalose was found to enhance viral accumulation by promoting post-attachment processes, including entry, shedding, and infectivity, despite no strong colocalization with the DENV envelope protein. Trehalose also increased lipid accumulation, hinting at links between trehalose metabolism, lipid biosynthesis, and autophagy. Crucially, trehalase inhibition curtailed both cell proliferation and DENV replication, highlighting its dual advantage: limiting mosquito development and reducing viral load. Overall, this study provides fresh insights into trehalose metabolism and mosquito-virus interactions while introducing a robust in vitro system for trehalase inhibitor screening with potential future applications in vector control strategies.

Reviewer #2: The manuscript by Andrew D. Marten et al. investigates the effects of trehalose metabolism in mosquito cells. They found that trehalose promotes cell growth and enhances DENV infectivity. However, I have several concerns that need to be addressed:

Reviewer #3: This study examined the role of trehalose in Aedes cell growth and dengue virus replication. The study is interesting, but one concern is that many of the observed phenotypes could be linked to differences in the build up of waste produced (specifically as growth differences are noted much later).

1. There is a wide literature base for the role of trehalose in relation to the trehalose metabolism, including knockdown studies in mosquitoes and fruit flies (see https://doi.org/10.1038/s41598-018-24893-z,
https://www.nature.com/articles/s42003-020-0889-1,
https://www.mdpi.com/2075-4450/13/11/1070). This includes studies directly on using trehalase inhibitors in control. The authors discuss some studies, but this is a much more vast field that needs coverage of more closely related studies.

2. Line 81-88 – This entire paragraph has little importance and should be deleted.

3. Most of the differences in cell growth seem to occur after day 5. Is it possible that trehalose is metabolized slower and can be used over longer periods? A steady nutrient release over extended periods could promote more growth

4. Statistical description as inadequate. There is no discussion of normality, etc. Also, there is just a general lack of description of how the data was analyzed. T-tests are likely not the most appropriate test in some cases.

5. Results – line 298: The description does not match the figure

6. Figure 2C and D. AS cell growth increased, I’d expect there would be a decline in glucose.

7. Line 335 – this is the circulating blood sugar and it is expected that enzymes involved in trehalose expression are expressed in many tissues.

8. Line 471 – As trehalose was the only factor that could serve as a substantial carbon source, the cells could be moving to a more quiescent state. As such, viral production may be reduced as Aag2 cells are no longer growing and replicating. These studies should likely be conducted in the presence of glucose (maybe 25%), which may confirm if this is due to a lack of general cell metabolism or only suppression of trehalose breakdown.

9. The discussion is heavily repetitive to the results and needs to be edited to provide more

10. Sample sizes and replication are not clear in most sections.

11. For all box plot, I would suggest overlay of the specific points onto the plots.

12. References are not uniform

**Part II – Major Issues: Key Experiments Required for Acceptance**

Reviewer #1: (No Response)

Reviewer #2: The authors claim that trehalose promotes cell proliferation. Since trehalose is metabolized into glucose, the mechanism by which trehalose promotes cell growth needs to be discussed in more detail. This explanation is essential to understand the underlying biological processes involved.

While the authors show that trehalose promotes virus infection, the mechanism behind this effect remains unclear. Given the transcriptome data, a more thorough investigation into how trehalose regulates virus infectivity is needed to clarify this relationship.

Line 302-303 states: “Together, these data indicated that Aag2 TREH is a secreted form of the enzyme and exists in the extracellular environment.” Based on what is known—that the gene explored is a homologue of TERH and the predicted results of subcellular localization—it could be suggested that the protein encoded by this gene is likely secretory. However, there is no direct evidence in this study supporting that the protein is secretory. Furhter investigation is needed to confirm this.

All experiments in the study rely on trehalose or inhibitor treatment. To strengthen the findings, genetic modifications targeting specific genes, such as TREH, should also be performed to assess the direct impact of these modifications on the observed effects.

If feasible, the authors should include experiments that evaluate the influence of trehalose on virus infection in mosquitoes, in order to verify the in vitro data within a more biologically relevant context.

Reviewer #3: 8. Line 471 – As trehalose was the only factor that could serve as a substantial carbon source, the cells could be moving to a more quiescent state. As such, viral production may be reduced as Aag2 cells are no longer growing and replicating. These studies should likely be conducted in the presence of glucose (maybe 25%), which may confirm if this is due to a lack of general cell metabolism or only suppression of trehalose breakdown.

**Part III – Minor Issues: Editorial and Data Presentation Modifications**

Reviewer #1: Aspects that could be clarified or further improved from my perspective:

• Figure 1: A) Why is the accession number for TPP accompanied by a question mark? Please clarify. B) The legend could be more descriptive, providing clearer guidance on how to interpret this panel, for SignalP non-users.

• Lines 296–297: I was unable to locate TPS (AAEL006446) in Tables S1, S2, or S3 of the referenced paper (Reference 42). While I found “AAEL010684,” its associated function is listed as “probable histone-lysine N-methyltransferase Mes-4, transcript variant X2,” not TPP. Furthermore, according to Tables S1 and S2 of Reference 42, TPP corresponds to other accession numbers and not AAEL010684.

• Line 300: Including a screenshot from the PSORT Wolf analysis, similar to the presentation of SignalP 6.0 in Panel 1C, would greatly enhance clarity. Alternatively, this could be provided in the Supplementary Figures for additional reference.

• Line 302: A more conclusive experiment, such as isolating the supernatant from cells, adding it to a trehalose solution, and subsequently measuring the glucose content, would provide stronger experimental proof. Trehalase could be present within the cells, converting trehalose into glucose as a non-secreted protein, especially considering trehalose is reported to enter the cells via passive transport. Additionally, if antibodies are available, a western blot analysis could help clarify this point.

• Figure 2F: The presentation of this experiment is not clear to me. If I understand correctly, the overall goal is to compare the efficiency of trehalase inhibitors. ValA, an inhibitor of trehalase, was tested alongside analogues of trehalose. Could you clarify whether these analogues also act as inhibitors by competing with native trehalose? Additionally, is it known how ValA works?

• Line 342: In the RNAseq experiment, the upregulation and downregulation observed in the trehalose-containing medium are compared to which condition? Is it the glucose-containing medium?

• Line 343: You mention that TREH expression did not change; however, I could not locate it in Supplementary Table 1. Is its absence from the table the reason for concluding that its expression did not change, or was it assessed elsewhere?

• Line 347: The statement regarding changes in genes involved in “the composition of the cell membrane [...] mitochondrial function” is unclear. It is not apparent where this information is extracted from, as it does not seem to be shown in Figures 3C or D. Could you provide a reference to the data supporting this statement?

• Line 348: A closing bracket appears to be missing.

• Line 353: The sentence is unclear. If trehalose is already provided in the cell medium, it seems unnecessary for the cells to express TPP to obtain trehalose. Could you clarify what is being tested in this context? Are you investigating whether Aag2 cells are capable of producing trehalose themselves?

• Line 366: The main conclusion of this section seems to be missing.

• Line 451: If the goal is to assess autophagy, why were autophagy markers, such as Lysotracker, not tested on the cells with and without trehalose? Including such markers would strengthen the conclusions.

• Line 454: Could you also test blocking autophagy in these cells when trehalose is provided? This would help determine whether the higher DENV levels are promoted by autophagy. For example, chloroquine could be used to inhibit autophagy.

• Line 483: This section appears to lack a clear conclusion.

• Line 511: The statement “Aag2 cells were not derived from fat body cells or tissue” raises a question. Since these cells originate from embryos, wouldn’t it be expected that some could differentiate into fat body-like cells and potentially excrete Tret1?

• Line 514: Which experiment could be performed to test this hypothesis? Would using Tret1 RNAi be a suitable approach?

• Line 517: Could you speculate on why this effect is also observed, to some extent, when glucose is provided?

• Line 566: How does the freeze-thaw event relate to virus replication in this context? Could you clarify its relevance?

• Figure 9E, F: When considering the number of cells, is the level of virus per cell consistent between the glucose and trehalose conditions?

• Line 582: You suggest that vertical transmission may not play a significant role. It is worth noting that a recent study by Julien Pompon demonstrated larval transmission through contamination with adult feces in Culex mosquitoes infected with West Nile virus. This highlights that such mechanisms could contribute to virus maintenance in the field.

• Lines 586–588: The sentence is too long, and should be splitted into two sentences.

• Line 656: A period is missing after “RNA.”

• Statistical significance is missing or unclear across several figures. In Figure 2, the significance of the difference at 7 dpi in panel B is not indicated, and statistical analyses are missing for all panels except A. In Figure 3B, no statistical analyses are provided to support the results. In Figure 4C, statistical analysis bars are absent despite a decrease being observed in Panel E for the two lower glucose concentrations, as mentioned in the text. Including statistical significance throughout these figures would strengthen the clarity and robustness of the findings.

I found the manuscript to be well-organized, clearly written, and supported by robust data. While there are areas that could benefit from further clarification or enhancement, they do not significantly detract from the overall strength of the study. It is important to emphasise that I believe this manuscript does not require additional experimental data to be accepted for publication.

That said, the conclusions could be further strengthened by demonstrating in vivo in Aedes aegypti that ValA not only prevents mosquito flight but also reduces DENV levels and prevents its excretion in saliva. While not essential for this study, which focuses on developing an in vitro tool to test trehalase inhibitors, this could be an interesting direction for future research. I would be open to collaborating on such experiments if the authors are interested.

In addition, I would find it interesting to explore whether the presence of trehalose or its conversion into glucose via trehalase leads to an increase in lipids, which consequently supports DENV replication. Using ValA on cells to observe a potential decrease in lipid levels or blocking lipid production to assess its impact on DENV replication could provide valuable insights. Again, while not required, these experiments could further enhance the manuscript’s impact.

Reviewer #2: In Fig. 2B&E, the annotations of different shapes should be included directly in the figure, rather than in the figure legends, to improve clarity and accessibility.

Lines 345-348 mention that differentially regulated genes are enriched in cell membrane processes, lipid biosynthesis, etc. However, the annotations in Fig. 3C&D are inconsistent with the description provided in the text. The figure should be revised to accurately reflect the described enrichments.

Reviewer #3: See summary

PLOS authors have the option to publish the peer review history of their article (what does this mean? ). If published, this will include your full peer review and any attached files.

**Do you want your identity to be public for this peer review?** For information about this choice, including consent withdrawal, please see our Privacy Policy .

Reviewer #1: **Yes: ** Faustine Ryckebusch

Reviewer #2: No

Reviewer #3: No

**Figure resubmission:**
---

## [Decision Letter · Decision Letter 1]

21 Apr 2025

Dear Dr. Conway,

We are pleased to inform you that your manuscript 'Trehalose supports the growth of Aedes aegypti cells and modifies gene expression and dengue virus type 2 replication' has been provisionally accepted for publication in PLOS Pathogens.

Best regards,

Alan G. Goodman

Academic Editor

PLOS Pathogens

Alexander Gorbalenya

Section Editor

PLOS Pathogens

Sumita Bhaduri-McIntosh

Editor-in-Chief

PLOS Pathogens

orcid.org/0000-0003-2946-9497

Michael Malim

Editor-in-Chief

PLOS Pathogens

orcid.org/0000-0002-7699-2064

Thank you for addressing the reviewers' previous concerns. Reviewer 1 notes two minor comments/edits that can be addressed during the typesetting phase. Congrats to all of the authors on their study.

Reviewer Comments (if any, and for reference):

Reviewer's Responses to Questions

**Part I - Summary**

Reviewer #1: I appreciate the efforts made by the authors to address the concerns I previously raised. In particular, the inclusion of data showing the presence of trehalase activity in the cell supernatant strengthens the claim that the enzyme is secreted by Aag2 cells. This addition supports the hypothesis that the trehalase involved is likely the secreted isoform, Trehalase 1 (Treh1), reinforcing the overall interpretation. Moreover, the manuscript is now clearer in linking the results to the corresponding figures, which improves the readability. The statistics have also been added to the figures as requested. In conclusion, i believe that the manuscript is now ready to be published.

Reviewer #2: (No Response)

Reviewer #3: Comments have been addressed

**Part II – Major Issues: Key Experiments Required for Acceptance**

Reviewer #1: (No Response)

Reviewer #2: (No Response)

Reviewer #3: (No Response)

**Part III – Minor Issues: Editorial and Data Presentation Modifications**

Reviewer #1: Although I previously suggested testing the involvement of autophagy, and I understand from the authors that such experiments can be technically challenging, I still find the current mention of autophagy in the results problematic. While trehalose has been linked to autophagy in mammalian cells, this has not been shown in insect cells. Since the RNAseq data do not support autophagy and no direct assays were performed, I recommend removing references to autophagy from the results. The focus should remain on lipid metabolism, with autophagy mentioned in the discussion as a possible, but unproven, mechanism.

The title could be improved to make it clearer and more focused. Right now, it feels a bit heavy because of the two "and"s. I also suggest highlighting the effect on the virus more clearly, since changes in metabolism are expected when sugar uptake or viral infection are involved. Here is a suggestion for the title:

– Trehalose Enhances (or Supports) Aedes aegypti Cell Growth and Modulates Dengue Virus 2 Replication

Finally, L645 “Aegypti” should not be capitalized.

Reviewer #2: (No Response)

Reviewer #3: (No Response)

PLOS authors have the option to publish the peer review history of their article (what does this mean? ). If published, this will include your full peer review and any attached files.

**Do you want your identity to be public for this peer review?** For information about this choice, including consent withdrawal, please see our Privacy Policy .

Reviewer #1: **Yes: ** Faustine Ryckebusch

Reviewer #2: No

Reviewer #3: No

---

## [Editor Report · Acceptance letter]

Dear Dr. Conway,

We are delighted to inform you that your manuscript, "Trehalose supports the growth of Aedes aegypti cells and modifies gene expression and dengue virus type 2 replication," has been formally accepted for publication in PLOS Pathogens.

Best regards,

Sumita Bhaduri-McIntosh

Editor-in-Chief

PLOS Pathogens

orcid.org/0000-0003-2946-9497

Michael Malim

Editor-in-Chief

PLOS Pathogens

orcid.org/0000-0002-7699-2064